# Feshbach hypothesis of high-Tc superconductivity in cuprates

**Lukas Homeier** [1,2] ✉, **Hannah Lange** [1,2,3], **Eugene Demler** [4],
**Annabelle Bohrdt** [2,5] **& Fabian Grusdt** [1,2] ✉

Resonant interactions associated with the emergence of a bound state constitute one of the cornerstones of modern many-body physics. Here we present a Feshbach perspective on the origin of strong pairing in Fermi-Hubbard type models. We perform a theoretical analysis of interactions between spin-polaron charge carriers in doped Mott insulators, modeled by a near-resonant two-channel scattering problem, and report evidence for Feshbach-type interactions in the $d_{x^2-y^2}$ channel, consistent with the established phenomenology of cuprates. Existing experimental and numerical results on hole-doped cuprates lead us to conjecture the existence of a light, long-lived, low-energy excited state of two holes, which enables near-resonant interactions. To put our theory to a test we suggest to use coincidence angle-resolved photoemission spectroscopy (cARPES), pair-tunneling measurements or pump-probe experiments. The emergent Feshbach resonance among spin-polarons could also underlie superconductivity in other doped anti-ferromagnetic Mott insulators highlighting its potential as a unifying strong-coupling pairing mechanism rooted in quantum magnetism.

High-temperature superconductivity in the cuprate compounds has a long and storied history and has remained an active field of research since its discovery in 1986[1]. The phase diagram of hole-doped cuprates features several types of orders or regimes, such as antiferromagnetism, the pseudogap phase, and the charge and spin density wave states[2–5]. How the superconducting phase arises from these potential parent states—as a competing or intertwined order or as a separate phenomenon—remains debated until today, and a generally accepted unifying theme capturing all phases continues to appear out of sight.

Naturally, the question about the origin of high-temperature superconductivity is closely tied to the question about the underlying pairing mechanism. In this regard, many aspects are well understood today. While in principle present, electron-phonon coupling responsible for conventional Bardeen–Cooper–Schrieffer (BCS) superconductivity[6] is widely believed to be insufficient and too weak for explaining the high transition temperatures and rich

phenomenology of cuprate compounds[7]. There is wide agreement that magnetic correlations play a central role in pairing, a view that is also supported by experimental evidence[8]. In the weak-coupling limit it is well understood how magnetic fluctuations can mediate $d$-wave attractive interactions[9,10], explaining the BCS-like appearance of the order parameter[11] in overdoped cuprates and its experimentally established nodal $d_{x^2-y^2}$ symmetry[12]. This picture of magnetic pairing has been further corroborated by numerous theoretical studies[9,10,13,14], leading to various proposed scenarios such as (para-) magnon exchange[15–18], spin-bag mechanism[19,20] and antiferromagnetic (AFM) Fermi liquids[21–23].

The strong coupling regime of the Fermi-Hubbard type models has been analyzed with numerical methods[24–31], variational RVB approach[2,32], by the functional renormalization group[10,14], and using field theoretical methods[33–36]. While these studies indicated the emergent strong pairing from magnetic fluctuations, they did not

[1]Department of Physics and Arnold Sommerfeld Center for Theoretical Physics (ASC), Ludwig-Maximilians-Universität München, München, Germany. [2]Munich Center for Quantum Science and Technology (MCQST), München, Germany. [3]Max-Planck-Institute for Quantum Optics, Garching, Germany. [4]Institute for Theoretical Physics, ETH Zurich, Zürich, Switzerland. [5]University of Regensburg, Regensburg, Germany. ✉e-mail: lukas.homeier@physik.uni-muenchen.de; fabian.grusdt@physik.uni-muenchen.de

provide a simple physical picture that explains its origin. To provide a simple physical explanation of this phenomenon is the central goal of our paper.

On the experimental side, a key motivation for our analysis comes from observations that the superconducting transition in underdoped cuprates is qualitatively different from the conventional, mean-field, BCS-like characteristics observed at high doping[11]. Uemura[37], Emery and Kivelson[38] suggested that the superconducting $T_c$ corresponds to the phase ordering transition with a non-BCS character[39–42]; thereby, holes may bind into (preformed) Cooper pairs already at temperatures higher than $T_c$. This points towards much stronger attractive interactions in the underdoped regime.

One of the surprising findings of recent state-of-the-art numerical studies in the context of Fermi-Hubbard-type models has been a strong sensitivity of $d$-wave pairing to details of the model. The ground state of the plain-vanilla Hubbard model at typical parameters has been shown to be non-superconducting[27] owing to a competition with stripes; however slight modifications of e.g., the band structure lead to $d$-wave superconductivity[31], in line with predictions by functional renormalization group calculations[10,14].

Another line of theoretical studies into the properties of high-$T_c$ cuprates has been to introduce strong attraction between charge carriers phenomenologically and investigate generic properties of such systems. This approach has been pursued by Nozières and Schmitt-Rink[43], Leggett[44], Randeria[45], and several others. Their methodology has been to introduce an effective attractive interaction between electrons that could be varied to tune the system between the BCS and BEC regimes, where spatially extended Cooper pairs continuously evolve into tightly bound bosonic pairs. These studies shed light on some of the surprising properties of high-$T_c$ cuprates[46,47] but do not address the origin of the strong attraction between electrons.

Here we propose an alternative perspective on the pairing mechanism at strong coupling, also based on magnetic correlations: emergent Feshbach resonance. The central message of this paper is that the microscopic mechanism behind strong pairing in doped Mott insulators is a Feshbach resonance that enhances interactions between doped holes. Recent numerical studies of the $t$-$J$ model[48] pointed out the existence of light but tightly bound, bosonic $d$-wave pairs of two holes with bi-polaron character in Mott insulators. These pairs complement individual fermionic holes to provide the basis of a two-channel Feshbach resonance model of pairing, see Fig. 1. We point out

that our scenario does not require the $d$-wave bi-polarons to be the ground state in systems with only two doped holes. Instead, we consider the case when the bi-polaron corresponds to a resonance at finite positive energy, however this energy is small. Then coupling of fermionic holes to the bosonic channel can give rise to strong attractive $d$-wave interactions between holes. In the scenario we describe, the short-distance microscopic structure of the fermionic and bosonic charge carriers in the AFM background leads to the long-lived state at finite energies associated with the bosonic bi-polaron. This distinguishes our perspective from geometric resonances usually discussed in the context of BEC-BCS crossover[43–45], where the bosonic state ceases to exist on the BCS side.

Feshbach resonances have been observed in various settings, from nuclear reactions in high-energy physics[49], cold atoms[50–52], to solid state materials[53,54]; they have recently been proposed as a route to unconventional pairing in strongly correlated electron systems[55–62], and give rise to universal strong-coupling behavior e.g., in the above-mentioned BEC-BCS crossover[46,47]. The microscopic description we provide here suggests that this scenario is also relevant to underdoped cuprates, and the Feshbach-paradigm naturally leads to near-resonant $d_{x^2-y^2}$ attraction, in line with the established phenomenology in these systems.

Our starting point is a weakly hole-doped parent state with strong local antiferromagnetism and a sufficiently large AFM correlation length $\xi_{AFM} \gg a$, where $a = 1$ is the lattice spacing. In cuprates, this regime is believed to be realized below the pseudogap temperature, $T < T^*$[14,26]. Charge carriers in this regime correspond to mobile holes and give rise to hole pockets, smoothly developing into Fermi-arcs[63], observed around the nodal points, $\mathbf{k} = (\pm\pi/2, \pm\pi/2)$, in angle-resolved photoemission spectroscopy (ARPES)[64] and by quantum oscillations[64,65]. At low doping in the square lattice Hubbard model with large on-site interaction $U \gg t$, the individual holes are well described by magnetic polarons, as revealed by analytical[66–71] and numerical studies[72–79], as well as experiments in solids[80] and recently in ultracold atoms[81].

In contrast, the two-hole excitation spectrum of the doped AFM is much harder to access experimentally and less understood theoretically. On the one hand, weak attraction between magnetic polarons, such as phonon- or magnon-exchange could suggest, could give rise to loosely bound Cooper-like pairs and would naturally lead to a BCS-type instability. On the other hand, numerical studies based on

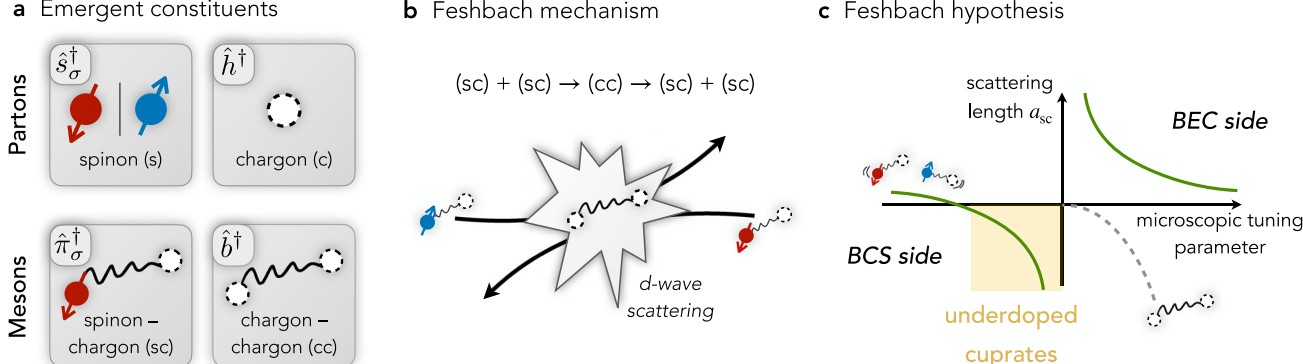

**Fig. 1 | Emergent Feshbach resonance. a** In the presence of strong AFM correlations, the emergent charge carriers can be described by partons, the spinon (s) and chargon (c), which form the meson bound states: fermionic (sc)'s and bosonic (cc)'s. If low-energy excitations in the ground state retain their fermionic character —as observed experimentally[63,64,144,145]—the bosonic two-hole pairs must correspond to meta-stable excited states. This situation is familiar from the case of Feshbach resonances and in the context of two particles: on the BCS side, the tightly bound bosonic state is replaced by a low energy two-particle resonance. **b** In the low-

energy scattering process between two (sc)'s with opposite spin, the mesons virtually recombine into an excited, light and tightly-bound (cc) bound state. Due to the mesons' internal structure, this process is dominated by $d_{x^2-y^2}$-wave scattering. **c** If the formation of a (cc) state is a resonant process, the scattering length diverges and yields strong effective interactions between the underlying (sc) mesons. In the Feshbach hypothesis, we conjecture that the normal state of underdoped cuprates is constituted by fermionic (sc)'s, located on the BCS side and in the vicinity of the scattering resonance.

Hubbard and $t$-$J$ models have indicated the existence of much more tightly bound pairs of holes[68,82–86] with only small energy differences between different pairing channels[48,84,87]. Moreover, recent density matrix renormalization group (DMRG) studies of the $t$-$J$ model reported the existence of long-lived two-hole resonances with distinct dispersion relations associated with different pairing symmetries[48]. The observed spectral features in that study can be explained by models of tightly bound holes connected with a string of displaced spins[83,88] (light bi-polarons).

## Results

### Feshbach hypothesis

To formulate our idea of Feshbach resonances, we need to describe in a common theoretical framework the individual fermionic charge carriers, which we model as magnetic polarons, and the tightly bound bosonic pairs. To this end, the parton picture provides a powerful construction, in which the fundamental constituents are decomposed into spinons (s) and chargons (c)[89–92], see Fig. 1a. Within this picture, magnetic polarons constitute a mesonic spinon-chargon bound state (sc)[93,94] with a rich set of internal ro-vibrational excitations. Hereby, the parton picture of confined (sc) bound states provides an alternative and fully consistent description of the single-hole quasiparticle observed in ARPES[94]. Similarly, the tightly-bound two-hole states are viewed as mesonic chargon-chargon bound states (cc). Their internal rotational quantum numbers correspond to different pairing symmetries[48,83,88]. Among the evidence for the parton picture are numerical studies at low doping which have confirmed the predicted internal excitations[94–98] and effective masses of the mesons[48,93].

Next, we discuss how the (sc) pairs (magnetic polarons) constituting the normal state in our model interact upon including a coupling to (cc) states. To this end, we consider an individual scattering event between two (sc) mesons with opposite spin. When they start to overlap spatially, they can recombine into a virtual (cc) state, realizing the sequence

$$(sc)_\uparrow + (sc)_\downarrow \rightarrow (cc) \rightarrow (sc)_\uparrow + (sc)_\downarrow. \tag{1}$$

This naturally leads us to a two-channel description of the emergent (mesonic) Feshbach resonance, with (sc)$^2$ defining the open- and (cc) defining the closed channel, respectively, see Fig. 1b. As we will argue below, the scattering process is dominated by a resonant $d$-wave (cc) state in the doped Hubbard model.

This leads us to formulate the following hypothesis:

- Cuprates, as well as commonly used models of the latter, i.e., Fermi-Hubbard or $t$-$J$ models at strong coupling, have a low-doping ground state close to a $d$-wave (sc)$^2$-(cc) scattering resonance.
- More specifically, underdoped cuprates are on the BCS side of the conjectured $d$-wave resonance, but sufficiently close for the induced attraction to overcome the intrinsic repulsion of two charge carriers in the (sc) channel.

This situation is illustrated in Fig. 1c, where we took the liberty to include an effective microscopic parameter tuning the relative energy of (sc) and (cc) channels independently. In reality, changing microscopic model parameters always affects the structure of both (sc) and (cc) mesons. On the level of theoretical models, a nearest-neighbor (NN) Hubbard interaction $V$ is a promising candidate to tune across the resonance[58,99–104].

Such tuning of the interactions may find a realization in solids when the screening of Coulomb interactions becomes increasingly poor in the low-doping regime[105]: Hence, at extremely low dopings, strong Coulomb repulsion is expected to lead to highly excited (cc) states and a normal state well in the BCS regime. Here, the attraction induced by couplings to (cc) states is expected to be unable to

overcome the intrinsic repulsion between the individual dopants, caused e.g., by kinetic (Pauli repulsion) or Coulomb effects. As the Coulomb repulsion becomes screened, the emergent Feshbach resonance is approached. Once the induced attraction among (sc)'s is sufficiently large, a weak-coupling $d$-wave BCS state of magnetic polarons is expected to form. This is consistent with recent observations of BCS quasiparticle peaks in ARPES studies of strongly underdoped layered $Ba_2Ca_5Cu_6O_{12}(F, O)_2$ compounds[64] at 4.3% hole doping. At higher doping, screening effects become even more effective. Here, the weak coupling BCS description can break down if near-resonant induced interactions are realized, which can lead to high critical temperatures $T_c$ as in BCS-BEC cross-over scenarios[46,47] and may explain the observed non-BCS nature of the superconducting transition[11].

How close a given cuprate compound, or state in a given Hubbard model, is to the conjectured resonance can depend sensitively on details. On the one hand, this may require some fine-tuning in order to reach the largest values of $T_c$ by reaching parameters closest to resonant interactions. On the other hand, as explained below, we expect relatively strong coupling between the open and closed channels, set by the super-exchange energy $J$ or the next-nearest neighbor (NNN) tunneling $t'$. This can lead to a broad Feshbach resonance, realizing strong attraction between (sc)'s even well before the resonance is reached[106]. Overall this scenario is broadly consistent with the considerable range of maximally achievable critical temperatures $T_c$ in different compounds.

### Effective string model

To support our hypothesis, we directly calculate the two-hole scattering interactions using a truncated basis approach in which we treat both the (sc) and (cc) channels on equal footing. To be concrete, we consider the $t$-$t'$-$J$ model[107],

$$
\begin{aligned}
\hat{\mathcal{H}}_{t-t'-J} = &-t \sum_{\langle \mathbf{i},\mathbf{j} \rangle} \sum_\sigma \hat{\mathcal{P}}\left(\hat{c}^\dagger_{\mathbf{i},\sigma}\hat{c}_{\mathbf{j},\sigma} + \text{h.c.}\right)\hat{\mathcal{P}} \\
&-t' \sum_{\langle\langle \mathbf{i},\mathbf{j} \rangle\rangle} \sum_\sigma \hat{\mathcal{P}}\left(\hat{c}^\dagger_{\mathbf{i},\sigma}\hat{c}_{\mathbf{j},\sigma} + \text{h.c.}\right)\hat{\mathcal{P}} \\
&+J_z \sum_{\langle \mathbf{i},\mathbf{j} \rangle}\left(\hat{S}^z_\mathbf{i}\hat{S}^z_\mathbf{j} - \frac{1}{4}\hat{n}_\mathbf{i}\hat{n}_\mathbf{j}\right) \\
&+\frac{J_\perp}{2}\sum_{\langle \mathbf{i},\mathbf{j} \rangle}\left(\hat{S}^+_\mathbf{i}\hat{S}^-_\mathbf{j} + \text{h.c.}\right),
\end{aligned}
\tag{2}
$$

with spin-1/2 fermions $\hat{c}^\dagger_{\mathbf{j},\sigma}$ residing on sites $\mathbf{j}$ with spin $\sigma = \downarrow, \uparrow$, and number (spin) operator $\hat{n}_\mathbf{j} = \hat{n}_{\mathbf{j},\downarrow} + \hat{n}_{\mathbf{j},\uparrow}$ ($\hat{\mathbf{S}}_\mathbf{j}$). We denote the links between NN and NNN sites as $\langle \mathbf{i},\mathbf{j} \rangle$ and $\langle\langle \mathbf{i},\mathbf{j} \rangle\rangle$, respectively. The Gutzwiller projector $\hat{\mathcal{P}}$ removes energetically costly double occupancies such that $\hat{n}_\mathbf{j} \leq 1$ for all $\mathbf{j}$ is enforced. The ground state $|0\rangle$ at half-filling is AFM Néel ordered.

The Feshbach hypothesis we introduce is conveniently formulated in the parton picture[93], which is an alternative framework and equivalent to a strong-coupling description of the underlying $\hat{c}_{\mathbf{j},\sigma}$ electrons; particularly single-hole ARPES spectra can be consistently understood in the parton picture[94] (see Methods). A related case is spin-charge separation in the one-dimensional Fermi-Hubbard model naturally described by the deconfined spinon (s) and chargon (c) parton excitations[108]; alternatively a formulation with $\hat{c}_{\mathbf{j},\sigma}$ fermions and non-local string operators could be applied, but it is more challenging to interpret the physical phenomenology. In contrast to the deconfined partons in one dimension, the antiferromagnetic correlations in the two-dimensional square lattice[14,26] lead to a confining string potential between the partons[66,67,93] and thus composite (sc) and (cc) bound states[88,109].

Next, we describe the (sc) and (cc) bound states, which arise in the sectors of the Hilbert space with one and two holes, respectively, upon doping the vacuum $|0\rangle$. We employ the semi-analytic geometric string

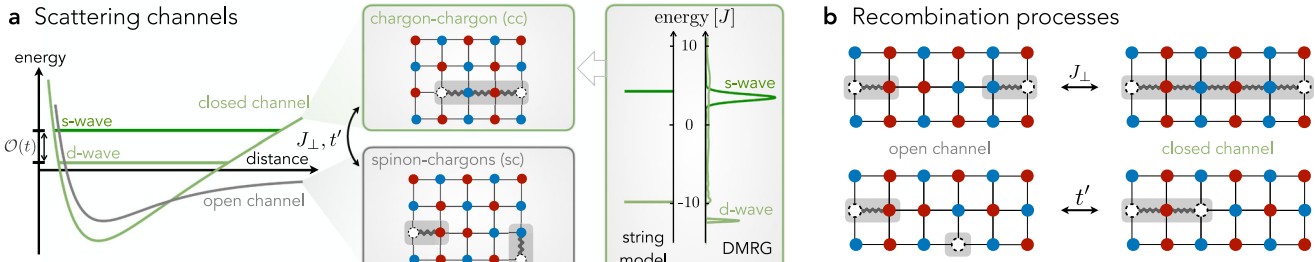

**Fig. 2 | Geometric string theory of the two-channel model. a** Magnetic polarons, i.e., (sc) mesons, are fermionic quasiparticles of single holes doped into an AFM Mott insulator. In contrast, tightly bound (cc) mesons exist as long-lived *s*- and *d*-wave resonances in the spectrum. The two types of mesons constitute open and closed scattering channels, and can be described in a truncated string basis defined in a classical Néel background (see insets); they are composed of chargons (white circles) connected to a spinon (bottom) or another chargon (top) at the opposite end of the string. The potential curves of the open and closed scattering channels

are sketches to illustrate the analogy to the Feshbach mechanism familiar from atomic physics. We plot the (cc) excitation spectrum at $\mathbf{Q} = \mathbf{0}$ obtained from the string model[88] (left) and from DMRG studies in the *t-J* model[48] (right), which shows the large energy separation between the *s*-wave and *d*-wave (cc) resonances. **b** Two spinons, each bound to a single chargon, can recombine into a longer string connecting the two chargons. Thereby spin exchange $J_\perp$ and NNN tunneling $t'$ processes couple the open and closed channel states and mediate an effective scattering interaction between the former.

theory, in which (sc) [(cc)] meson eigenstates are expanded in a truncated basis of orthonormal string states $|\mathbf{j}_\sigma, \Sigma\rangle$ [$|\mathbf{x}_c, \Sigma_{cc}\rangle$]. Here $\mathbf{j}_\sigma$ [$\mathbf{x}_c$] denotes the spinon [first chargon] position and $\Sigma$ [$\Sigma_{cc}$] is a sequence of non-retracing string segments connecting to the other parton, providing a confining force. These states can be constructed explicitly for one and two holes in a classical Néel state by displacing spins along the string $\Sigma$, see Figure 2a and Methods, and have been shown to provide an accurate, yet analytically tractable, approximation of the mesons' ground and excited states[66,67,83,87,88,97,109].

The open channel (sc) states constitute the starting point of the analysis. The gap to their internal excitations is sufficiently large $\mathcal{O}(J)$ such that at low energies only s-wave (sc) mesons $\hat{\pi}_{\mathbf{k},\sigma}$ exist[98]. Therefore, in the low-doping limit, we describe the open channel by the following free-fermion Hamiltonian,

$$\hat{\mathcal{H}}_{\text{open}} = \sum_{\mathbf{k},\sigma} \left[ \varepsilon_{sc}(\mathbf{k}) - \mu \right] \hat{\pi}^\dagger_{\mathbf{k},\sigma} \hat{\pi}_{\mathbf{k},\sigma}. \tag{3}$$

The dispersion relation $\varepsilon_{sc}(\mathbf{k})$ has its minimum at the nodal points $\mathbf{k} = (\pm\pi/2, \pm\pi/2)$[69,70], around which two Fermi pockets form upon increasing the chemical potential $\mu$[64]. In particular, the dispersion relation (see Methods) is extracted from numerical studies of the single hole[110] and extended heuristically to finite doping, which we expect to be valid description at low doping due to the qualitatively good agreement between the shape of the hole pockets and the single-hole dispersion. Moreover, since we assume the AFM correlation length $\xi$ to be sufficiently larger than the size of a (sc) meson, we can analyze the case of an AFM-ordered state. Thus, all excitations are defined with momenta $\mathbf{k}$ in the magnetic Brillouin zone (MBZ), which is rotated by $\pi/4$ and reduced with respect to the original crystal Brillouin zone (CBZ).

The closed channel states are constituted by (cc) mesons $\hat{b}^\dagger_{\mathbf{Q},\alpha,m_4}$ with momentum $\mathbf{Q}$ and band index $\alpha$[48,88]. As in the case of (sc) mesons we ignore vibrational excitations; it is however important to include their rotational structure described by the $C_4$ angular momentum $m_4 = 0, \ldots, 3$. The resulting free (cc) Hamiltonian is

$$\hat{\mathcal{H}}_{\text{closed}} = \sum_{\mathbf{Q},\alpha,m_4} \varepsilon^{m_4}_{cc}(\mathbf{Q}) \hat{b}^\dagger_{\mathbf{Q},\alpha,m_4} \hat{b}_{\mathbf{Q},\alpha,m_4}, \tag{4}$$

with the dispersion $\varepsilon^{m_4}_{cc}(\mathbf{Q}) = -\frac{2}{3}J(\cos(Q_x) + \cos(Q_y) - 2) + \Delta E_{m_4}$ of the tightly bound pair[48]; here $\Delta E_{m_4}$ denotes the energy above the scattering threshold. In the truncated string basis, the (cc) states do not distinguish between the two magnetic sublattices and can be defined

in the CBZ. In order to describe their coupling to the open channel in the next step, we fold the CBZ into the MBZ by restricting ourselves to $\mathbf{Q} \in$ MBZ and introducing the band index $\alpha = 0, 1$. Because the (cc) wavefunction of the two fermionic holes can be (anti)symmetric in the band index, any value of $m_4$ can be realized at the $C_4$-invariant momenta; in particular at $\mathbf{Q} = \mathbf{0}$ shown below to be relevant for low-energy scattering.

Now we turn to a description of the coupling between the open and closed channels. In the effective string basis, we recognize microscopic processes $\propto J_\perp, t'$ leading to recombination processes from a (sc)² into a (cc) configuration, see Figure 2b: We highlight that the recombination processes only require local terms in the Hamiltonian acting on states with AFM correlations of a few lattice sites. That this scenario could be realized in the underdoped cuprates is corroborated by numerical cluster dynamical mean-field studies[111] and experiments with ultracold fermions in optical lattices[112,113] both finding AFM and spin-polaron correlations beyond 5% doping. Our assumption is further consistent with fractionalized Fermi liquids[33]: recent studies using a variational spin liquid ansatz for the normal state find polaronic correlation across the underdoped regime[114,115]. This suggests that the Feshbach mechanism is a valid description in the underdoped cuprates and can be established even for normal states without long-range AFM order, which host (sc) and (cc) constituents.

In order to describe the corresponding low-energy two-particle scattering problem, we consider two (sc)'s in a spin-singlet state and with momenta $\pm\mathbf{k}$ from opposite sides of the Fermi surface. Note that we focus here on intra-pocket recombinations: two (sc)'s from the same hole pocket recombine into a (cc) pair with total momentum $\mathbf{Q} = \boldsymbol{\pi}$, which leads to zero-momentum Cooper pairs in the MBZ. Hence the allowed pairing symmetries are of *d*- or *s*-wave nature.

To understand into which (cc) states the (sc)² can recombine, we formulate the relevant selection rules associated with the translational and $C_4$-rotational symmetries of the system. The former ensures that total momentum is conserved, i.e., only couplings to (cc) mesons with $C_4$-invariant momenta $\mathbf{Q} = \mathbf{0}$ or $\boldsymbol{\pi}$ are allowed. At these momenta, the latter symmetry further ensures conservation of total $C_4$ angular momentum $m_4$. Since the individual (sc)'s have s-wave (internal) character, only the orbital angular momentum associated with their relative motion contributes to $m_4$, as defined by the corresponding zero-momentum Cooper pair operator,

$$\hat{\Delta}^\dagger_{m_4} = \frac{1}{\sqrt{2}} \sum_{\mathbf{k}} f_{m_4}(\mathbf{k}) \left( \hat{\pi}^\dagger_{-\mathbf{k},\uparrow} \hat{\pi}^\dagger_{\mathbf{k},\downarrow} - \hat{\pi}^\dagger_{-\mathbf{k},\downarrow} \hat{\pi}^\dagger_{\mathbf{k},\uparrow} \right). \tag{5}$$

Here, the function $f_{m_4}(\mathbf{k})$ transforms as $f_{m_4}(\mathbf{k}) \rightarrow e^{-im_4\pi/2} f_{m_4}(\mathbf{k})$ under $C_4$ rotations. Since only $m_4 = 0$ (s-wave) and $m_4 = 2$ (d-wave) lead to non-vanishing $\hat{\Delta}_{m_4}^\dagger$, we conclude from the derived selection rules that only couplings to closed channel (cc) states with $d$- or $s$-wave symmetry are allowed. In principle, our formalism allows us to include finite momentum Cooper pairs relevant for inter-pocket scattering and potentially relevant for going beyond BCS mean-field theory.

In the next step, we integrate out the closed (cc) channels, which yields mediated interactions that we can describe by the effective Hamiltonian

$$\hat{\mathcal{H}}_{\text{int}} = \sum_{\mathbf{k},\mathbf{k}'} V_{\mathbf{k},\mathbf{k}'} \hat{\pi}_{-\mathbf{k},\uparrow}^\dagger \hat{\pi}_{\mathbf{k},\downarrow}^\dagger \hat{\pi}_{-\mathbf{k}',\uparrow} \hat{\pi}_{\mathbf{k}',\downarrow}, \qquad (6)$$

with scattering matrix elements $V_{\mathbf{k},\mathbf{k}'}$ of the form

$$V_{\mathbf{k},\mathbf{k}'} = \frac{1}{L^2} \sum_{m_4 = 0,2} \frac{1}{\Delta E_{m_4}} \mathcal{M}_{m_4}^*(\mathbf{k}) \mathcal{M}_{m_4}(\mathbf{k}'). \qquad (7)$$

Here we sum over the two allowed closed channels, $m_4 = 0, 2$. The form factors $\mathcal{M}_{m_4}(\mathbf{k})$ will be calculated below from matrix elements coupling open and closed channels that we obtain within the effective string model. $L^2$ denotes the two-dimensional volume.

An emergent Feshbach resonance with divergent attractive interactions is realized when the closed channel approaches the scattering threshold from above, i.e., $\Delta E \rightarrow 0^+$. At the considered (cc) momentum $\mathbf{Q} = \mathbf{0}$, only the $d$-wave (cc) state has low energy $\Delta E_2$; in contrast, the s-wave (cc) state has energy $\Delta E_0 = \mathcal{O}(t)$ owing to its strong center-of-mass dispersion. This follows from string model calculations[83,88] and has been confirmed by large-scale exact-diagonalization[85] and DMRG studies[48] in the $t$-$J$ model. We show the corresponding energy distribution curves of the pair-spectral functions at $\mathbf{Q} = \mathbf{0}$ in Fig. 2a (right column), where the large splitting is clearly visible.

This leads to the important conclusion that low-energy scattering of magnetic polarons is dominated by couplings to the $d$-wave (cc) channel; namely, since $\Delta E_0 \approx t$, couplings to the $s$-wave channel can be safely neglected in $V_{\mathbf{k},\mathbf{k}'}$, Eq. (7). This justifies the simplified two-channel model underlying the Feshbach hypothesis formulated earlier in our article. Accurate numerical calculations of the (cc) gap $\Delta E_2$ are extremely challenging: since it is defined as the distance of the bare (cc) energy from the two-particle scattering threshold, it constitutes a small difference of two much larger quantities and becomes sensitive to details. Nevertheless we note that numerous studies have concluded that tightly-bound $d$-wave pairs, i.e., with (cc) character, exist at low energies close to the scattering threshold, see e.g., refs. [48,85,87,88]. This justifies our conjecture that strongly interacting doped Hubbard models at low doping are close to an emergent $d$-wave Feshbach resonance.

### Scattering interaction

Now we use the truncated string basis to calculate the form factors $\mathcal{M}_2(\mathbf{k})$ characterizing the open-closed channel coupling. From the underlying $t - t' - J$ model we find (see[116]) two contributions corresponding to the different recombination processes illustrated in Fig. 2b,

$$\mathcal{M}_2(\mathbf{k}) = J_\perp \mathcal{M}_2^{J_\perp}(\mathbf{k}) - \text{sgn}(\delta) |t'| \mathcal{M}_2^{t'}(\mathbf{k}). \qquad (8)$$

This result assumes perturbative $|t'| \ll t$, such that the effective string model remains valid. Moreover we generalized our description to include hole and electron doping $\delta$, characterized by $\text{sgn}(\delta) = +1$ (holes) and $\text{sgn}(\delta) = -1$ (electrons) respectively; in deriving Eq. (8) we further assumed $t' < 0$ in Eq. (2). A typical value used to model cuprates is $t'/|t| = -0.2$[117,118]. In the following, we extend the two-body

scattering description to a many-body problem of weakly interacting magnetic polarons and discuss implications of the effective model at finite doping.

We show the two contributions to the form factor separately in Fig. 3a, b and find a rich momentum dependence with a sign structure reflecting the underlying $d$-wave symmetry of (cc) mesons. Notably, within a weak coupling BCS description of the (sc) Fermi sea, the superconducting order parameter $\Delta(\mathbf{k}) \propto \mathcal{M}_2(\mathbf{k})$ follows directly from the BCS gap equation for Eq. (6), see ref. [116]. Therefore, Fig. 3 reveals the structure of the pairing gap. In particular, we find for both recombination processes that the form factor (and hence the pairing gap) is dominated by the ubiquitous $d_{x^2-y^2}$ nodal structure observed in hole doped cuprate compounds.

Beyond the $d_{x^2-y^2}$ structure dominating around the hole pockets we find additional nodal lines in the CBZ, see Fig. 3a, originating from a superimposed extended s-wave structure of the interactions. This rich momentum-dependence of the induced interactions reflects the extended spatial structure of the closed-channel (cc) state mediating the attractive interaction. Nevertheless, within our theory, combining $J_\perp$ and $t'$ processes, pairing is dominated by $d_{x^2-y^2}$ structure in the entire low hole doping regime $\delta < 15\%$, with only small deviations[116] from the ideal $\Delta_{x^2-y^2}(\mathbf{k}) \simeq |\cos(k_x) - \cos(k_y)|$ structure; therefore the additional nodal lines are outside the immediately relevant experimental regime. In fact, the gap structure we find is consistent with ARPES measurements of the pairing gap in cuprate compounds[64,119,120], which indicate that the quasiparticle gap differs from the simple $d$-wave form $\Delta(\mathbf{k}) \propto \cos(2\phi_{\mathbf{k}})$ by the addition of a small $\delta\Delta(\mathbf{k}) \propto \cos(6\phi_{\mathbf{k}})$ component (see Methods).

### Experimental signatures

Now we turn to possible experimental signatures of the Feshbach hypothesis. The key ingredient of the proposed pairing mechanism is the existence of a tightly bound (cc) channel at low excitation energies above the Fermi energy. At this point, it is important to contrast the (cc) mesons to pre-formed Cooper pairs expected above the superconducting critical $T_c$ where phase-coherence rather than pairing disappears at low doping[37,38]. From our perspective, pre-formed Cooper pairs should be of (sc)²-type and exist just above $T_c$, whereas (cc) pairs can exist separately at higher energies and potentially well above $T_c$.

While the (cc) channel gives rise to a doping-dependent ARPES feature in single-particle spectroscopy, we find the signal to be very weak (see "Methods" and ref. [116]). Therefore, we turn our attention to two-hole (correlation) spectroscopy, where numerical studies[48,85] found the pronounced (cc) peaks shown in Fig. 2a (right column). A direct measurement of such two-hole spectra requires removal of a tightly-bound pair, with or without well-defined $C_4$ angular momentum, at a well-defined energy $\omega$ and momentum $\mathbf{k}$. Theoretical proposals have been made on how this can be achieved in coincidence ARPES spectroscopy, in a one-photon-in-two-electron-out[121,122] or a two-photon-in-two-electron-out[123] configuration. While the former needs high-energy photons, the latter requires low-intensity light in order to distinguish the coincidence signal from the one-particle ARPES background[124], making both challenging. In Fig. 4a, we illustrate the correlated ARPES process, which removes adjacent electrons and creates a hole pair having overlap with all angular momentum $m_4$ channels of the (cc) bound state if $\mathbf{k} \neq \mathbf{0}$. The energy onset of the signal relative to the Fermi energy $E_F$ and at $\mathbf{Q} = \mathbf{0}$ would directly measure the detuning $\Delta E_{m_4}$ from the (cc) resonance in our proposed Feshbach scenario.

Further, we suggest two potential experiments based on the coherent tunneling of pairs. First, if a sample with $T > T_c$ is brought in contact with a superconductor, the Cooper pairs from the latter can tunnel through the junction into the (cc) channel, similar to Anderson–Goldman pair tunneling[125]. Such experiments have already revealed some signatures of enhanced pairing fluctuations above $T_c$[41].

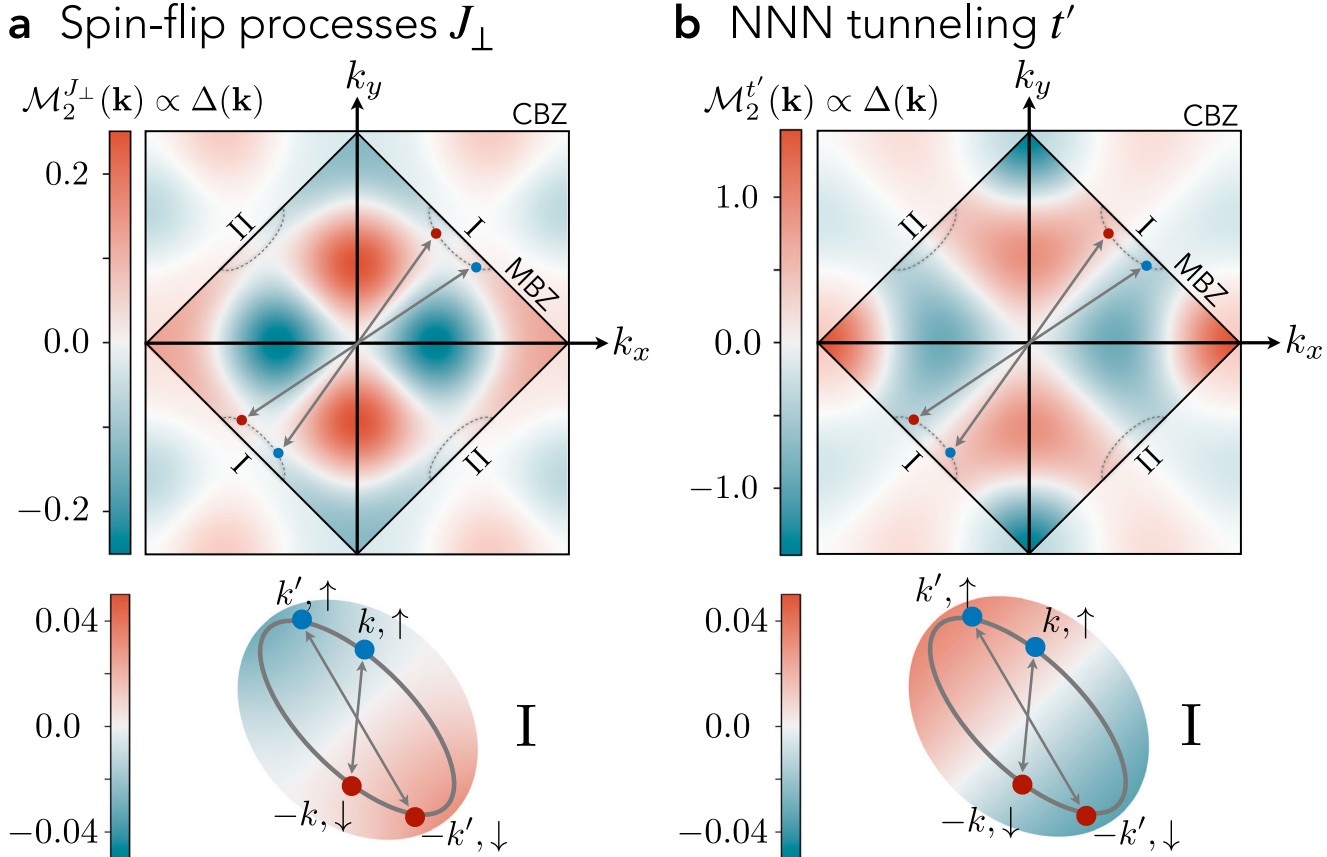

**Fig. 3 | Symmetry of the pairing interaction.** We plot the dimensionless open-closed channel form factors $\mathcal{M}^{J_\perp}(\mathbf{k})$ from spin-flip processes **a**, and $\mathcal{M}^{t'}(\mathbf{k})$ from NNN tunneling processes **b**, as calculated within the two-channel model of the emergent Feshbach resonance. They directly reflect the sign structure of the resulting superconducting order parameter $\Delta(\mathbf{k})$, and show a strong momentum dependence. Outside the MBZ we used reduced opacity for clarity. Upon doping, charge carriers fill up the two hole pockets I and II, realizing scattering of total momentum $\mathbf{Q} = \mathbf{0}$ pairs on the Fermi surfaces (bottom). The mediated interactions, and hence the pairing gap, vanish along the white nodal lines, featuring a dominant $d_{x^2-y^2}$ nodal structure in the CBZ.

By adding an in-plane magnetic field and a voltage across the junction, the pair spectra could be directly probed with combined energy and momentum resolution[48,126]. Second, scanning tunneling noise spectroscopy can directly probe local pairs by analyzing autocorrelations of the current fluctuations[127,128]. We expect an onset of enhanced pairing fluctuations at an energy $\Delta E_{m_4}$ above $E_F$ as a direct signature of the near-resonant (cc) state, which should be present even above $T > T_c$. This is in line with recent noise-spectroscopy experiments performed on LSCO[42] which reported an onset of enhanced pairing fluctuations around an energy scale of 10 meV.

Finally, we propose to verify the existence of the tightly-bound (cc) state in pump-probe experiments, in which the properties of the transient state are modified, see Fig. 4b. By off-resonantly driving a Raman-active phonon mode, the microscopic parameters $t$, $t'$ and $J$ can be modulated in time by the pump pulse, with frequency $\omega$. From our microscopic model we predict that this can drive the transition between the open (sc)² and closed channel (cc), achieving tunable interactions in the transient state which become resonant for $\omega = \Delta E_{m_4}$. On resonance we can expect creating a large population of long-lived metastable (cc) pairs. This transient state can exhibit optical properties similar to systems with strong superconducting correlations. This scenario is potentially relevant for understanding light-induced superconductivity[129].

**Feshbach resonances in other settings**

The Feshbach scenario[130,131] we suggest may provide a long-sought common perspective on pairing in a whole range of strongly correlated systems. Experiments in solid-state systems have demonstrated that in fine-tuned settings resonant Feshbach interactions can lead to bound states of excitons[53,54]. In the past years, resonant pairing interactions have been proposed as a path towards unconventional superconductivity in correlated materials and synthetic quantum matter, such as two-dimensional semiconductors[55–57] and transition metal dichalcogenides[60–62], which may allow to engineer materials with topological superconductivity[57,61]. The observation of high-temperature superconductivity in the pressurized bilayer nickelates further suggests the possibility of an emergent two-channel model constituted by charge-1e spinful fermions and tightly-bound charge-2e bosons[58,59,132] in these materials. Numerical studies in one-dimensional toy models have found a pairing dome in analogy to the phenomenology in the cuprate superconductors[58].

Our study of doped antiferromagnets can therefore provide a microscopic perspective towards a unifying paradigm of unconventional superconductivity in two-dimensional materials originating from repulsive interactions. We identify the existence of a near-resonant, bosonic (cc) channel as a key ingredient in our model. The experimental probes for solids we propose allow one to search for direct signatures of such charge-2e pairs and therefore to test for Feshbach scenarios in cuprates and other strongly-correlated materials.

## Discussion

To summarize, we propose a new perspective on the pairing mechanism potentially underlying high-temperature superconductivity as observed in the cuprate compounds. It is based on the

## a Coincidence ARPES

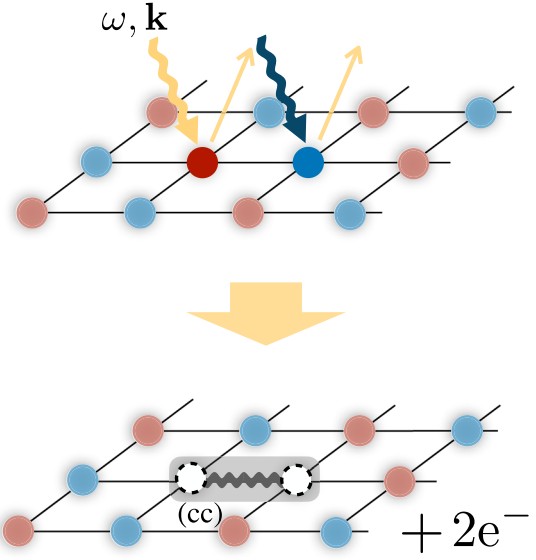

## b Pump-probe scheme

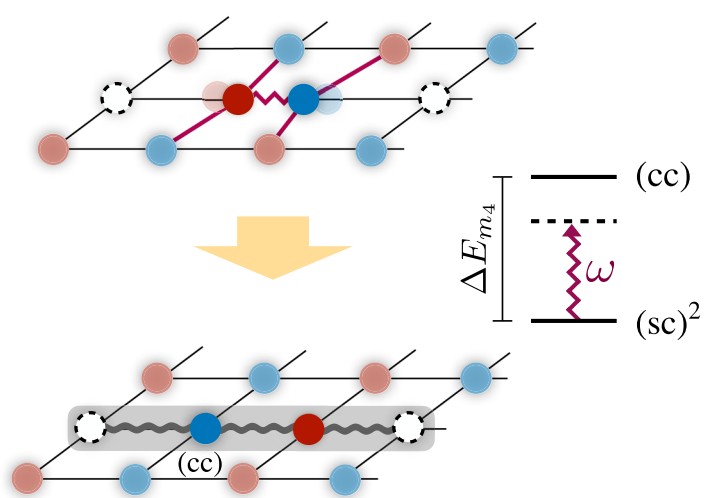

**Fig. 4 | Experimental signatures of the closed channel (cc) state. a** Two photons with energy and momentum ($\omega$, **k**) remove two adjacent electrons from the system, which can be detected in the correlations of the photoelectrons (coincidence ARPES). This gives direct access to the dispersion relation of the (cc) meson as shown for **Q** = 0 in Figure 2a (right). **b** We propose a pump-probe protocol, in which

a Raman active phonon-mode is driven at frequency $\omega$ (purple). This modulates the microscopic couplings and drives a transition between the open channel (sc)$^2$ and closed channel (cc) states such that the energy difference $\Delta E_{m_4}$ in the proposed Feshbach model can be overcome, potentially allowing to reach a transient state with resonant interactions.

idea that an emergent Feshbach resonance between magnetic polaron-like constituents, with spinon-chargon character, of the low-doping normal state may arise when a near-resonant tightly bound bi-polaronic state of two holes exists at low excitation energies $\Delta E_2$. We find that the parton picture is a natural description for the two-channel model, which allows us to perform calculations and derive the effective $d_{x^2-y^2}$-wave pairing interaction between charge carriers.

A direct application of our theory is to analyze electron-doped cuprates, where the topology of the magnetic polaron Fermi surface changes[133] and the realized pairing symmetry remains subject of debate. Although in our calculations we assumed a long-range ordered AFM, our ideas directly extend to disordered normal states such as fractionalized Fermi liquids[33,134–137]. Particularly, the proposed Feshbach mechanism only requires short-range polaronic correlations, see Fig. 2, for which indications are found across the underdoped regime in ultracold atom experiments[112,113,138] consistent with alternative descriptions of the normal state[114,115] and cluster dynamical mean-field theory[111].

In addition to the possible experimental signature discussed above, numerical studies as well as quantum simulation experiments[108] provide further possibilities to test our hypothesis. On one hand, advances in calculating one- and two-particle spectral functions numerically[139] will allow to search for more direct signatures of the emergent Feshbach resonance. On the other hand, ultracold atoms in optical lattices[108] or tweezer arrays[140] allow to study clean systems with widely tunable parameters, in and out-of-equilibrium, and including higher-order correlation functions[141]. A possible application of the latter would be to look for direct signatures of the strings defining the structure of spinon-chargon and chargon-chargon pairs, in a wide range of dopings.

## Methods
### Parton picture
The proposed Feshbach hypothesis is formulated within a parton picture starting from spinons (s) and chargons (c) constituents. The

purpose of this section is to provide a pedagogical overview of parton models in doped AFMs.

The motion of a single hole through an AFM background has been studied well before the discovery of high-Tc superconductivity in the cuprate compounds. The phenomenology of a confining potential from displaced spins in an AFM background, see Fig. 2a (inset), was already described by Bulaevski, Nagaev and Khomskii in 1968. They recognized that this confining force plays a crucial role in the dispersion of the hole. Later, the displaced spins were described as string objects and a formal framework was developed[67,68,72,83], however, the notion of partons was not yet introduced.

With the discovery of high-Tc superconductivity, numerical and theoretical studies of the single hole in an AFM were performed, see e.g., refs. 69,70,110, which predict (i) the dispersion minimum of the hole at **k** = ($\pi$/2, $\pi$/2) consistent with ARPES measurements in cuprates[63] and (ii) long-lived quasiparticle excitation peaks at higher energies. The connection between the excitation spectrum to the string excitations was established by Liu and Manousakis[79].

The numerical studies find a strongly renormalized dispersion of the hole $\propto J$, while other properties such as the Drude conductivity are related to the tunneling $\propto t$[93,142]. Beran, Poilblanc and Lauglin proposed in ref. 93 that a picture of composite partons, where a spinon (s) and chargon (c) are confined by a string-like object, naturally explains the observed properties: if the composite particle is constituted by a heavy spinon with mass $\propto 1/J$ and a light chargon with mass $\propto 1/t$, it is consistent that properties of the quasiparticle can depend on either the spinon or chargon. Further, the string excitations found numerically are interpreted in this picture as internal excitations of the parton bound state[93,109].

In our article, the parton picture provides us with a powerful framework: indeed, describing the open and closed scattering channel in terms of two distinct parton bound states enables us to initially develop the Feshbach hypothesis. We note that a description using the underlying $\hat{c}_{j,\sigma}$ electrons would be fully consistent with the parton picture but requires the introduction of non-local, multi-body string

operators. As we describe next, a much more accessible and computationally tractable approach is to introduce string objects on the level of the Hilbert space with spinons (s) and chargons (c) attached to the end of the strings[109,143].

## Geometric string theory

In the proposed Feshbach scenario two meson-like parton-bound states virtually recombine from an open channel (sc)$^2$ into the closed channel (cc). Hereby, the spatial structure of the meson-bound states is crucial to determine the properties of the scattering channels and recombination processes. A semi-analytical description of the bound state wavefunction for $t \gg J$ can be obtained in a truncated basis (geometric string) approach[88,97,109] spanning the basis states of the open and closed channel Hilbert spaces, $\mathscr{H} = \mathscr{H}_{\text{open}} \oplus \mathscr{H}_{\text{closed}}$, with exactly two holes. The strategy is to determine the properties of the emergent Feshbach resonance in the few-body setting of two holes, for which we can apply the geometric string theory to derive the effective interactions $V_{\mathbf{k}, \mathbf{k}'}$, see Eq. (6). Since the effective interactions describe the leading order two-body interactions between magnetic polarons, it allows us to elevate the few-body problem to a many-body Hamiltonian in the low-doping regime with weakly interacting (sc)$^2$s/magnetic polarons.

The truncated basis states in the open channel are given by $\mathscr{H}_{\text{open}} = \{|\mathbf{j}_\downarrow, \Sigma_\downarrow \rangle \otimes |\mathbf{j}_\uparrow, \Sigma_\uparrow \rangle\}$ and are defined as follows. Starting from the undoped AFM Néel background, holes are inserted at position $\mathbf{j}_\sigma$ ($\sigma = \downarrow, \uparrow$) but not on adjacent sites; further the fermionic two hole states have to be antisymmetrized. In the limit $t \gg J$, we assume the spin background to be frozen such that the individual holes can tunnel on NN sites and disturb the AFM spin background. The path of the holes, excluding self-retracing paths, is encoded in the string $\Sigma_\sigma$ with length $\ell_\sigma$ and thus the string object contains information about the displacement of the AFM ordered background. Further, the string connects the spinon $\sigma$ at position $\mathbf{j}_\sigma$ with the corresponding chargon, see Fig. 2a (inset). For distant holes, i.e., for the asymptotic free states in the scattering problem, the strings give rise to a linear confining potential $V_\Sigma \propto J_z \cdot \ell_\sigma$ resulting in parton bound states with a discrete spectrum; these asymptotic parton wavefunctions correspond to the wavefunctions obtained for single holes[109,143]. Due to the increasing potential energy with the length of the strings, the basis states can be truncated in a controlled way.

A good parametrization of the internal states is given by the string length $\ell_\sigma$ and the angles between string segments of length $\ell_\sigma$ and $\ell_\sigma + 1$; thus, the eigenstates can be labeled by vibrational and rotational quantum numbers, respectively, which become particularly good quantum numbers around $C_4$ invariant momenta $\mathbf{k}_\sigma$. The internal ground state of these bound states are the magnetic polarons $\hat{n}_\sigma^\dagger$ or (sc) meson-bound states in our model. Refined theoretical modeling and including the spin-flip terms $J_\perp$ can accurately describe the (sc) dispersion relation $\varepsilon_{\text{sc}}(\mathbf{k})$[97,109,143]. From numerical simulations[110], the form of the (sc) dispersion relation can be approximated by

$$\varepsilon_{\text{sc}}(\mathbf{k}) = A[\cos(2k_x) + \cos(2k_y)] \\ + B[\cos(k_x + k_y) + \cos(k_x - k_y)], \quad (9)$$

where $A$ and $B$ are fit parameters; for realistic parameters in cuprates compounds $t/J = 10/3$, the fit parameters are $A = 0.31$ and $B = 0.44$[110].

In our work, we use the dispersion relation obtained from a single hole and extend it to finite doping by introducing a chemical potential for the magnetic polarons. This is consistent with ARPES measurements at low doping[63]; recent experiments suggest that the single-hole dispersion smoothly crosses over into the Fermi arcs[64]. The geometric string theory provides an alternative formalism, which is fully consistent with single-hole ARPES[94]. In the string picture, the spectral weight corresponds to the weight of the bound state wavefunction at string length $\ell_\sigma = 0$; this property is derived from the definition of the

spectral weight in single-hole ARPES $Z_{\mathbf{k}, \sigma} \propto |\langle \psi^{N-1} | \hat{c}_{\mathbf{k}, \sigma} | \psi^N \rangle|^2$, where $|\psi^M\rangle$ is the ground state of a system with $M$ electrons[124], by comparing it to the length $\ell_\sigma = 0$ string basis state with $|\mathbf{k}, \ell = 0\rangle = \hat{c}_{\mathbf{k}, \sigma} |\psi^N\rangle$ and the (sc) bound state wavefunction $|\psi^{N-1}\rangle$ of the single hole.

Similarly, the truncated basis states for the closed channel are given by $\mathscr{H}_{\text{closed}} = \{|\mathbf{x}_c, \Sigma_{cc}\rangle\}$[83,88]. Here, the basis is constructed in the following way: we insert a pair of (distinguishable) holes on adjacent sites and label the site of the first hole by $\mathbf{x}_c$, which is connected by the string $\Sigma_{cc}$ of length $\ell_{cc}$ to the second hole. Now, both holes can tunnel while remaining connected by the string tracing their paths' through the spin background. The length of the string is associated with a confining potential. Again, at the C4-invariant momenta $\mathbf{Q} = \mathbf{0}, \boldsymbol{\pi}$ the quantum numbers of the bound state are given by the vibrational and rotational modes; the former are energetically gapped $\mathcal{O}(t^{1/3}J^{2/3})$ and we assume the vibrational ground state. Importantly, the eigenstates have to be antisymmetrized in order to describe fermionic two-hole bound states. In this procedure we assume a total spin singlet. Hence, the bound state does not carry any spinon degree-of-freedom and can be described by its chargon-chargon (cc) content. The (cc) bound states $\hat{b}_{\mathbf{Q}, \alpha, m_4}^\dagger$ can have different rotational eigenvalues $m_4 = 0, \ldots, 3$. In contrast to the spinon positions $\mathbf{j}_\sigma$ in the open channel, the position $\mathbf{x}_c$ is defined on the crystal lattice. Therefore, we can either (i) describe momenta in the CBZ or (ii) introduce a band index $\alpha$ that is acquired from band folding into the MBZ, see main text. The (cc) bound state has substantial overlap with a state, where two adjacent electrons are removed from an AFM Mott insulator (see section on cARPES) corresponding to a string length $\ell_{cc} = 1$ state in our description. This gives rise to an ARPES-type two-hole spectrum with spectral weight at energies $E$ as plotted in Fig. 2a (right) for $\mathbf{Q} = \mathbf{0}$ in the MBZ obtained from geometric string theory[88] in the $t$-$J_z$ model and state-of-the-art DMRG calculations of the $t$-$J$ model on $40 \times 4$ cylinders[48].

The bare energy difference between the open and closed channel states $\Delta E_{m_4}$ determines the distance in energy from the Feshbach resonance and, hence, controls the scattering length. The previous DMRG studies[48] have also determined the dispersion relations of the (cc) bound states revealing a strong $m_4$ dependence consistent with geometric string calculations[88]. While the s-wave (cc) state is highly mobile $\propto t$, the $d$-wave band is flat in the $t$-$J_z$ limit and only disperses by including spin flip-flop terms $J_\perp$. At the relevant scattering momenta $\mathbf{Q} = \mathbf{0}, \boldsymbol{\pi}$, the (cc)'s internal ground state has $d$-wave symmetry with gap $\mathcal{O}(t)$ to internal rotational excitations. Therefore, the near-resonant low-lying closed channel state is of $d$-wave symmetry with the (fitted) dispersion $\varepsilon_{cc}(\mathbf{k}) = -J(\cos(k_x) + \cos(k_y) - 2) + \Delta E_{m_4 = 2}$.

## Scattering form factors

The form factor $\mathcal{M}_{m_4}(\mathbf{k})$ is proportional to the coupling matrix elements between open (sc) and closed (cc) channel states. Specifically, we consider open-closed channel couplings given by

$$\hat{\mathcal{H}}_{\text{oc}} = \hat{\mathcal{H}}_{J_\perp} + \hat{\mathcal{H}}_{t'} \\ \hat{\mathcal{H}}_{J_\perp} = \frac{J_\perp}{2} \sum_{\langle \mathbf{i}, \mathbf{j} \rangle} \left( \hat{S}_i^+ \hat{S}_j^- + \text{H.c.} \right) \\ \hat{\mathcal{H}}_{t'} = -t' \sum_{\langle\langle \mathbf{i}, \mathbf{j} \rangle\rangle} \sum_\sigma \left( \hat{c}_{\mathbf{i}, \sigma}^\dagger \hat{c}_{\mathbf{j}, \sigma} + \text{H.c.} \right). \quad (10)$$

Note that we include the NNN tunneling $t'$ in a perturbative description, i.e., we assume the wavefunction and band structure of the (sc) and (cc) bound state for $t' = 0$ and calculate their coupling for small NNN tunneling $t'$. The coupling between and within the open and closed channels is illustrated in Fig. 5.

As argued in the main text, the zero momentum scattering ($\mathbf{Q} = \mathbf{0}$) is relevant for low-energy intra-pocket scattering of magnetic polarons. In the MBZ, this also allows for closed channel states with $\mathbf{Q} = \mathbf{0}$ and antisymmetric band index. Therefore, the relevant form factor is

related to the matrix elements by

$$\mathcal{M}_2(\mathbf{k}) = J_\perp \mathcal{M}_2^{J_\perp}(\mathbf{k}) + t' \mathcal{M}_2^{t'}(\mathbf{k}) \tag{11a}$$

$$\frac{\kappa}{\sqrt{L^2}} \mathcal{M}_2^\kappa(\mathbf{k}) = \langle 0 | \hat{b}_{\mathbf{Q}=\boldsymbol{\pi}, m_4=2} \hat{\mathcal{H}}_\kappa \hat{\pi}_{-\mathbf{k},\uparrow}^\dagger \hat{\pi}_{\mathbf{k},\downarrow}^\dagger | 0 \rangle, \tag{11b}$$

with $\kappa = J_\perp, t'$. These matrix elements can be used in a full BEC-BCS crossover description, which captures strong hybridization between the two channels.

Here, we integrate out the closed channel, i.e., the closed channel is only virtually occupied in the scattering process, yielding the scattering Eq. (7). We emphasize that the structure of the (cc) bound state is inherited through the coupling matrix elements. To calculate the matrix elements, we expand the mesons' bound state wavefunctions in

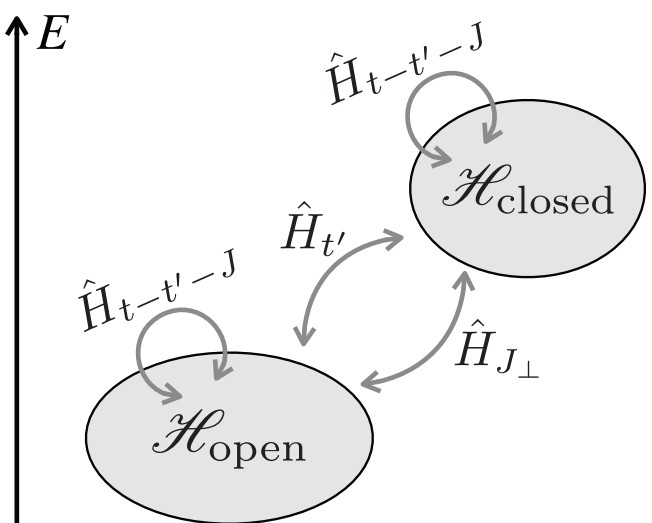

**Fig. 5 | Open and closed channel couplings.** The Hamiltonian we consider, Eq. (10), determines both (i) the properties of the bare scattering channels, such as the dispersion relations and the energy difference $\Delta E_{m_4}$, as well as (ii) the coupling between open and closed channel states by spin flip-flop $J_\perp$ and NNN tunneling $t'$ processes. In this work, we only consider small NNN tunneling $t'$ and do not include its effect on the dispersion relation or the (sc) bound state, which are found to be small in studies using a refined truncated basis method[116].

their string length. Since the bulk of the meson wavefunctions only span across a few lattice sites, the largest contribution to the form factors are given by short string length states, where two magnetic polarons of length $\ell_\sigma \approx 1$–2 recombine with a (cc) state of length $\ell_{cc} \approx 3$–4; the convergence of the form factor in the string length is confirmed in ref. 116. In Fig. 3, we show the resulting form factors for (cc) strings up to length $\ell_{cc} = 5$ using the meson wavefunctions for $t/J = 3$ determined in refs. 88,143.

The form factors $\mathcal{M}_2(\mathbf{k})$ we consider in our model are constituted by spin-flip $J_\perp$ and NNN tunneling $t'$ processes. Further, the string description gives rise to an effective sign difference between electron ($\delta < 0$) and hole ($\delta > 0$) doping, see Eq. (8), that originates from a broken particle-hole symmetry for $t' \neq 0$. In the main text in Fig. 3 we plot the two contributions $\mathcal{M}_2^{J_\perp}(\mathbf{k})$ and $\mathcal{M}_2^{t'}(\mathbf{k})$ separately.

In Figure 6, we additionally provide the combined form factors $\mathcal{M}_2(\mathbf{k})$ for various $t'/J_\perp$; note that we do not take into account the attributed change in the (sc) and (cc) wavefunctions by the NNN tunneling. We find that the nodal ring of the spin-flip form factor moves away from the hole pockets and towards the center of the Brillouin zone as $t'$ is increased. Further, we sketch the hole pockets for $t/J = 10/3$ and $\delta = 15\%$[110], which—for realistic values of $t'$ in cuprates—do not touch the nodal ring. Our findings are consistent with ARPES gap measurements in cuprate superconductors.

**Gap anisotropy**

In our phenomenological string model, we find additional extended s-wave structure, which occurs from beyond point-like interactions, i.e., from the finite extend of the meson wavefunctions. Similarly, precise measurements of the superconducting gap in underdoped Bi-2212 indicate an anisotropy in momentum space that cannot be explained by the plain vanilla $d_{x^2-y^2}$ symmetry as argued by Mesot et al.[119]. Instead, it was found that the function

$$\Delta(\phi_\mathbf{k}) \propto B \cos(2\phi_\mathbf{k}) + (1 - B) \cos(6\phi_\mathbf{k}) \tag{12}$$

captures the gap structure in underdoped samples, where $B$ is used as a fit parameter. In ARPES measurements the pairing gap is determined along the Fermi surface, conveniently parameterized by the angle $\phi_\mathbf{k}$, see Fig. 7b (inset) for definition. To model the cuprates in our calculations, we assume a Fermi surface of magnetic polarons for $t' = 0$[110] as shown in Fig. 7 (inset). Next, we calculate the BCS pairing gap $\Delta(\mathbf{k}) \propto \mathcal{M}_2(\mathbf{k})$ (see main text) for different parameters $|t'/J_\perp| = 0.05, 0.2$ and extract its features along the magnetic polaron's Fermi surface. To this end, we fit to (i) $\cos(2\phi_\mathbf{k})$ (i.e., $B = 1$) and (ii) to the refined gap function Eq. (12) with $B \in [0, 1]$. The fitted curves are shown by the dashed and

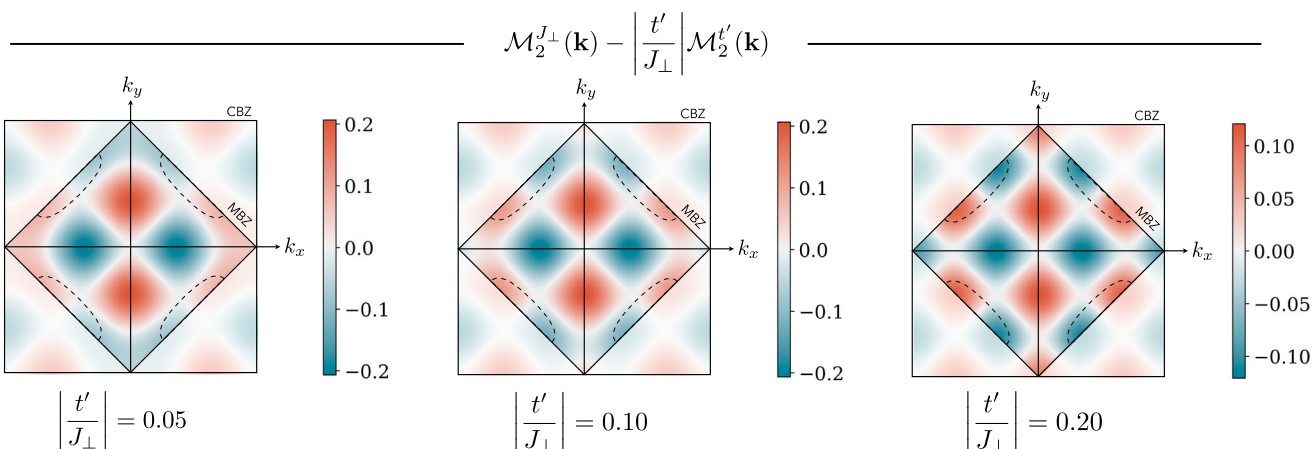

$$\mathcal{M}_2^{J_\perp}(\mathbf{k}) - \left| \frac{t'}{J_\perp} \right| \mathcal{M}_2^{t'}(\mathbf{k})$$

$$\left| \frac{t'}{J_\perp} \right| = 0.05 \qquad \left| \frac{t'}{J_\perp} \right| = 0.10 \qquad \left| \frac{t'}{J_\perp} \right| = 0.20$$

**Fig. 6 | Combined $t'$ and $J_\perp$ form factors.** We show the form factor $\mathcal{M}_2(\mathbf{k})$ for hole doping in cuprate superconductors ($t'/|t| < 0$) for $|t'/J_\perp| = 0.05, 0.1, 0.2$. The hole pockets (dashed lines) correspond to the single hole magnetic polaron dispersion with $t/J = 10/3$[110] at relatively large doping $\delta = 15\%$.

solid gray lines in Fig. 7. We find excellent agreement of our calculations to the fit for the second case (ii), consistent with ARPES measurements[119].

## Coincidence ARPES (cARPES)

Coincidence ARPES, where two correlated photoelectrons are emitted, gives access to the (cc) channel with different angular momentum $m_4$. This can be realized in both a one-photon-in-two-electron-out and a two-photon-in-two-electron-out scheme. The former requires high energy photons and therefore places an additional experimental challenge, while for the latter the typical ARPES photon energies can be used[124] and will be considered in the following.

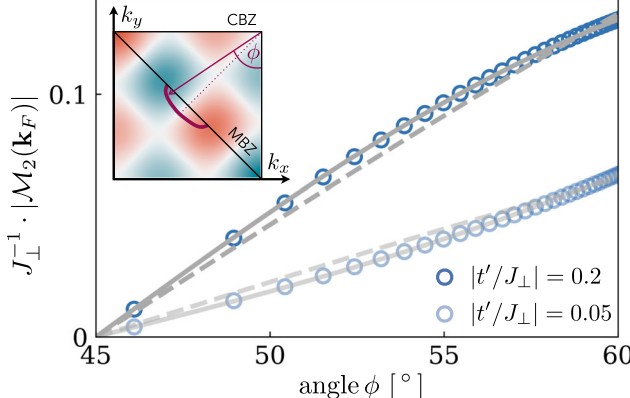

**Fig. 7 | Gap anisotropy.** The form factor $\mathcal{M}_2(\mathbf{k})$ is directly related to the pairing gap $\Delta(\mathbf{k})$. We assume an elliptical Fermi surface of magnetic polarons, i.e., hole pockets, at $\delta = 10\%$ doping. The Fermi surface is parameterized by the angle $\phi_\mathbf{k}$ (inset). The blue circles show the pairing gap, calculated from our two channel model, along the Fermi surface for exemplary values $|t'/J_\perp| = 0.05, 0.2$. We fit the pairing gap using a plain vanilla $d_{x^2-y^2}$ gap symmetry (dashed gray) and a refined gap function including $\cos 6\phi_\mathbf{k}$ terms (solid gray). The latter is an excellent fit function for an entire range of parameters $|t'/J_\perp|$.

The two in-coming photons have momentum $\mathbf{k}_1$ and $\mathbf{k}_2$ (out-going photoelectrons have momentum $\mathbf{k}_3$ and $\mathbf{k}_4$). Here, we consider the special case with $\mathbf{k}_1 = \mathbf{k}_2 \equiv \mathbf{k}$, and $\mathbf{k}_3 = \mathbf{k} + \delta\mathbf{k}$ and $\mathbf{k}_4 = \mathbf{k} - \delta\mathbf{k}$. In this sector, there is no momentum transfer to the sample and thus we probe (cc) states with $\mathbf{Q} = \mathbf{0}, \boldsymbol{\pi}$ in the MBZ.

In the main text, we argue that for $\delta\mathbf{k} \neq 0$ the coupling matrix elements or spectral weight of coincidence ARPES is non-zero for any angular momentum $m_4$. To show this, we consider the matrix elements $\mathcal{R}(\delta\mathbf{k})$ for removing two adjacent electrons at distance $\mathbf{r}_n$, given by four individual processes on the square lattice:

$$\mathcal{R}(\delta\mathbf{k}) \propto \sum_{n=0}^{3} e^{i\frac{\pi}{2}nm_4} e^{-i\mathbf{r}_n \cdot \delta\mathbf{k}}. \tag{13}$$

The matrix elements are momentum dependent ($\delta\mathbf{k}$) and admit the rotational symmetry $m_4$ of the (cc) channel. Since the low-energy channel has $d$-wave symmetry ($m_4 = 2$), the matrix elements vanish along the nodal directions, and in particular at $\delta\mathbf{k} = 0$. However, for $\delta\mathbf{k} \neq 0$ we find non-zero matrix elements to the (cc) bound state.

## Single-hole ARPES

Alternatively to coincidence ARPES, one can−in principle−obtain a less-direct signature of (cc) states in single-hole APRES spectroscopy. However we find the signal to be very weak, as we will show in the following. Nevertheless, high resolution ARPES in ultra-clean samples or quantum simulation may enable to measure signatures of the (cc) bound state in the future.

In order to extract two-hole properties from the ARPES spectrum, involving the removal of just one electron, we make use of the non-zero density of already existing charge carriers. In addition to the low-energy one-hole excitations of the normal state directly probed in ARPES, we predict a broad and doping-dependent feature associated with the low-energy tightly-bound (cc) pair. Namely, if the additional hole created by the photo-electron is in the vicinity of an existing magnetic polaron, i.e., (sc), this state is generically expected to overlap with the tightly bound (cc) pair, see Fig. 8. Owing to the two-body

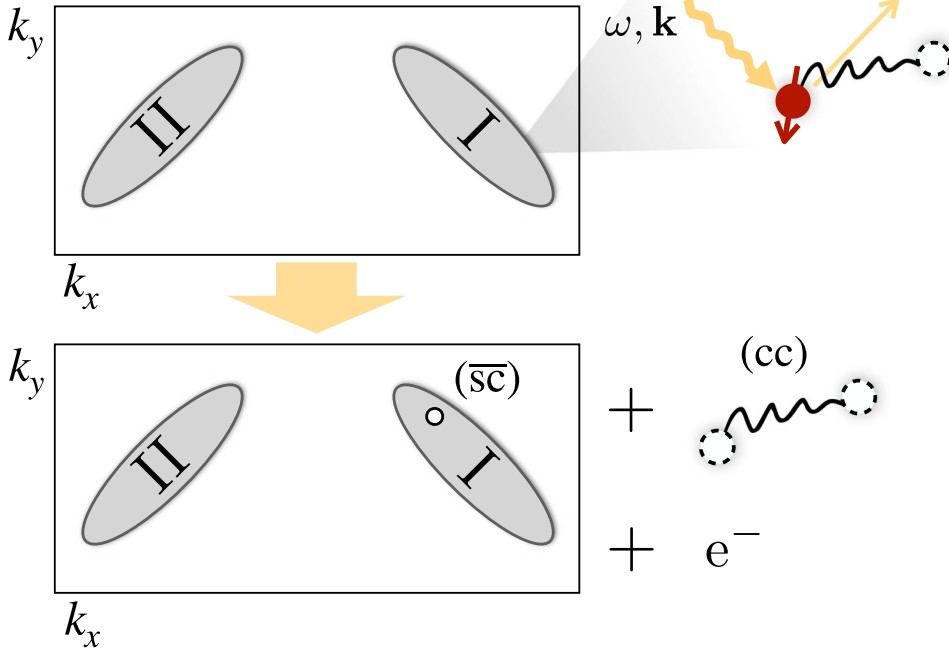

**Fig. 8 | Single-hole ARPES.** In the low-doping regime, the Fermi sea of charge carriers forms two hole pockets I and II[64]. We consider photoemission processes, where the photoelectron is removed in the vicinity of an already existing magnetic polaron. This process can couple to the (cc) channel by leaving behind a hole-type excitation ($\overline{\text{sc}}$) in the Fermi sea; the two-body nature gives rise to broad spectral continuum.

nature of the created excitation, constituted by the (cc) pair and a hole-type excitation $(\overline{sc})$ in the normal state, energy and momentum can be distributed and give rise to a two-body $(\overline{sc}) + (cc)$ continuum.

The shape of the broad $(\overline{sc}) + (cc)$ feature follows from the known dispersion relations $\varepsilon_{sc}(\mathbf{p})$ (see e.g., [110]) of the (sc) and $\varepsilon_{cc}(\mathbf{p}) = -J(\cos(p_x) + \cos(p_y) - 2) + \Delta E_{m_4}$[48] of the (cc) constituents. Using momentum conservation we predicted a contribution to the ARPES spectrum at momentum $\mathbf{k}$ and energy $\omega$,

$$
\begin{aligned}
A_{\overline{sc}+cc}(\mathbf{k}, \omega) = \int \frac{d^2\mathbf{p}}{(2\pi)^2}\, n^F(\varepsilon_{sc}(\mathbf{p}) - \mu)\, \mathcal{R}(\mathbf{k}, \mathbf{p}) \\
\times \delta(\omega + \varepsilon_{sc}(\mathbf{p}) - \mu - \varepsilon_{cc}(\mathbf{k} + \mathbf{p})),
\end{aligned} \tag{14}
$$

where $n^F(\varepsilon)$ is the Fermi-Dirac distribution and $\mathcal{R}(\mathbf{k}, \mathbf{p})$ includes matrix elements reflecting the microscopic structure of (sc) and (cc) constituents. The matrix elements $\mathcal{R}(\mathbf{k}, \mathbf{p})$ can be approximated using the truncated basis technique of the (sc) and (cc) channels.

Altogether, the combination of several factors lead to a very weak, estimated signal of about $10^{-3}$ relative to the quasiparticle peak at the Fermi energy: (i) The signal depends on the hole density, which is small in the regime of interest, (ii) the coupling matrix elements are suppressed for the momenta $\mathbf{k}$, which probe (cc) state at total momentum $\mathbf{Q} = 0, \pi$, and (iii) the dispersion relation of the (cc) channel yields a broad and flat signal distributing the spectral weight.

## Data availability

All data presented in this article have been obtained by the authors using their own code and are available within the main Article and Supplementary Information file.

## Code availability

Our calculations are explained in the methods section. Code is available upon request.

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

## Acknowledgements

We thank Pit Bermes, Roderich Moessner, Lode Pollet, Tizian Blatz, Timothy Harris, Immanuel Bloch, Markus Greiner, Ulrich Schollwöck, Atac Imamoglu, Assa Auerbach, Matteo Mitrano and Walter Metzner for fruitful discussions. This research was funded by the Deutsche Forschungsgemeinschaft (DFG, German Research Foundation) under Germany's Excellence Strategy—EXC-2111—390814868 and has received funding from the European Research Council (ERC) under the European Union's Horizon 2020 research and innovation program (Grant Agreement no 948141) - ERC Starting Grant SimUcQuam. L.H. was supported by the Studienstiftung des deutschen Volkes. H.L. acknowledges support from the International Max Planck Research School. E.D. received funding from the SNSF (project 200021_212899), the Swiss State Secretariat for Education, Research and Innovation (contract number UeM019-1), and the ARO grant W911NF-20-1-0163.

## Author contributions
L.H. performed all calculations. F.G. supervised the work. L.H., A.B., and F.G. devised the conceptual idea. All authors jointly analyzed the results and proposed possible experimental tests. L.H. and F.G. wrote the manuscript, with input from H.L., E.D., and A.B.

## Funding

## Competing interests
The authors declare no competing interests.
