## [Transparent Peer Review file · Nature Communications]

Feshbach hypothesis of high-T_c superconductivity in cuprates

Corresponding Author: Professor Fabian Grusdt

Version 0:

Reviewer comments:

Reviewer #1

(Remarks to the Author)

(Remarks on code availability)

Reviewer #2

(Remarks to the Author)

The manuscript by L. Homeier et al. touches on the old but still largely unresolved problem, i.e. on the origin of superconductivity in 2D Hubbard-like models. To this end, it uses a speculative hypothesis which postulates that a Feshbach resonance between the fermionic mesons and the bosonic two-hole states may explain the apparent pairing in this class of models. While certainly such studies are needed, and the concept in principle is interesting, I doubt whether this study can be published in Nature Communications. In general, I find the study, even for a "hypothesis-like" paper, way too speculative, since a number of reasoning elements are either entirely skipped or are not really clear. More precisely, I have the following issues with the presented study:

(I) Validity of the "sc" description:

(a) It is written that the "sc" fermionic meson is *the* quasiparticle that is observed in e.g. ARPES on cuprates or in the spectral function of the Hubbard / t-J models. However, such an "sc" bound state binds a hole (c) with a "neighbouring" spin (s) -- and in ARPES the quasiparticle response is actually widely-understood (cf. [69, 71, 95, 134]) as originating just from a single hole. Therefore, the Authors' approach goes against the current understanding and needs further (and substantiated) explanation.

(b) What are the explicit conditions that (3) is valid? What is the explicit equation for the dispersion relation in (3) and how it is obtained from the Hamiltonian (2)?

(II) Validity of the "cc" description:

(a) Papers [54] and [88] (that form the basis of the "cc" dynamics description) seem to contradict each other: according to [54] the tightly-bound pair disperses by $\sim t$ but according to [88] the tightly-bound pair disperses by J_{\perp} . Which one is correct and why?

(b) In the current manuscript it is assumed that "cc" moves due to J_{\perp} and paper [54] is cited there -- but, as just written above, paper [54] claims that "we find a long-lived, tightly bound state of two holes, which can move as fast as the hole hopping t ." (page 3 of [54]).

(c) I could not find the explicit equation for ϵ_{cc} in [54] so I would appreciate if the Authors explain ("step-by-

step") how it is obtained from the Hamiltonian (2).

(III) The resonance and the recombination process:

(a) The cartoon of Fig. 2(a), which shows the scattering channel energies as functions of the distance between the constituents, should be backed by calculations.

(b) The physical mechanisms behind the recombination process are not really clear / consistent:

-- On one hand, t' is not included in the description of "cc" and "sc" states (including t' would qualitatively affect their dispersion relations). Justification of this approximation should be given.

-- On the other hand, J_{\perp} is already crucial for the mobility of "cc" states so it is not clear to me how it can at the same time lead to the recombination process shown in Fig. 2(b). This double role of J_{\perp} should be explained.

(Remarks on code availability)

Reviewer #3

(Remarks to the Author)

This paper advances a theory of pairing in strongly correlated fermionic Hubbard-type models. The emergent spinon and chargeon entities, which respectively carry spin and charge and are fermionic, can bind into bosonic spinon-chargeon (sc) and chargeon-chargeon (cc) "mesons." The sc mesons undergo strong scattering and the Coulomb repulsion between them is apparently overcome by a Feshbach resonance between these emergent excitations.

This paper is very clear and well-written. The figures are all nicely done and the sketches are helpfully evocative. As it is not possible to show from first principles whether a given Hubbard-type model will result in the Feshbach resonance pairing physics for the notional emergent partons, the authors appropriately focus on describing experimental signatures of their scenario suggest experiments based on coherent tunneling of Cooper pairs as well as predicted signatures within ARPES.

Fundamental aspects of the Feshbach resonance pairing mechanism are also articulated in ref. 61, which describes a theory of pairing via Feshbach resonance in nickelate superconductors. However, there is much that is new in the current submission, which is specifically relevant to cuprates. There is only a faint relation to the work in Rocchina Caivano et al. (Supercond. Sci. Technol. 22, 014004 (2009)) and in the paper of Squire and March (Int. Jour. Quantum Chem. 110, 2808 (2010)).

Overall, the theory and suggested experimental probes are speculative. It is not clear to me whether any of the experimental signatures proposed could not also be present in a model which did not invoke the Feshbach resonance.

The theoretical work here is very strong indeed, but would be more compelling if accompanied by comparison with actual experimental data.

(Remarks on code availability)

Version 1:

Reviewer comments:

Reviewer #1

(Remarks to the Author)

What I wanted in a revised manuscript was not delivered. My request of the authors was to make a greater effort towards pedagogy and to provide a more useful paper for the readership-- not just in an afterthought fashion as a two paragraph "Perspectives" section, but in greater depth. For example, I do not believe many experimentalists, even those working in high temperature superconductivity, will be able to take away a simple physical picture or be inspired to do future experiments.

I realize that this Journal aims to publish "important advances of significance to specialists", but in my view the specialists here are not just theorists but experimentalists as well. And this is an exclusively theoretical and, indeed speculative paper and would have been much stronger had there been some degree of experimental support. Predictions for future experiments are not sufficient to address this concern.

On the positive side this paper is ambitious and in this regard I applaud the authors. In principle, it can set a new direction for theory-- particularly if there is more than numerical support to follow. And particularly, it it looks to have a broader relevance, going beyond a focus on the cuprates but addressing

other correlated superconductors where a generic Hubbard model appears relevant. Lacking this support at this time, I think it might be more suitable for a Journal such as Physical Review X.

For these reasons I will not recommend its publication in Nature Communications.

Reviewer #2

(Remarks to the Author)

First of all, I would like to thank the Authors for their detailed response to all my comments.

In general, I am quite satisfied to their replies to my comments I(a), II, and III: these replies have helped me much better understand the content of the paper and they have clarified some of my doubts. On these points, I just have one request: I would appreciate, if the replies to my concerns were more thoroughly and explicitly incorporated in the manuscript [in particular, Fig. 2(a) can stay as is but an explicit discussions of its shortcomings is needed, in my opinion].

However, reply to point I(b) makes me think that the whole discussion is only valid in an extreme regime of one / two holes -- which cannot be extended to the interesting (for the understanding of superconductivity in the cuprates or in the Hubbard model) regime of $> \sim 5\%$ doping. This is because the Authors assume that the "sc" bound states resemble the spin polarons, as known from the single-hole study. In fact, this assumption is central both to the form of Eq. (3) as well as to the derivation of the coupling between open and closed channel that follows from [Homeier et al., PRB 109, 125135 (2024)]. The latter statement is also clearly visible from cartoon in Fig. 2(b) in which the recombination process efficiently works, due to the spinon sitting not on the same site as the holon but rather next to it. On the other hand, already at *few percent doping* the dominant part of the ARPES spectrum of the Hubbard model is dominated by a "hole-like" quasiparticle and the spin polaron is strongly suppressed, cf. Fig. 1 (b) of [Kohnno, PRL 108, 076401 (2012)] or Fig. 3(b) of [Wang et al., Comm. Phys. 3, 210 (2020)]. So, in my opinion, the main question remains: Can the postulated mechanism (in particular, the coupling between closed and open channels) be at all valid once $I=0$ in the "sc" states?

Last but not least, I am also quite worried that this paper has too much overlap with a recent article published by some of the Authors as [Homeier et al., PRB 109, 125135 (2024)]. That is why I am still quite reluctant in recommending this manuscript for publication in Nature Communications.

Reviewer #3

(Remarks to the Author)

My original report was the least critical of the three. My main issue was regarding to what extent experimental data supports the conclusion that the Feshbach resonance mechanism provides the best interpretation of the data. In the authors' response they share some preliminary results comparing SCPA and DMRG results for the two-hole ARPES spectra. While interesting, I don't think this really answers my question, so the paper still strikes me as rather speculative. In that regard, I believe this manuscript might be more suitable for a journal such as Phys. Rev. B, although I would not object to publication in Nature. As I had noted before, it is quite clear and well-written and the figures are very well-done. Furthermore, there is significant new material here vis-a-vis earlier work on this mechanism as a driver of high temperature superconductivity.

Version 2:

Reviewer comments:

Reviewer #2

(Remarks to the Author)

First of all, I would like to thank the Authors for their very detailed reply to my latest report. In particular, I appreciate (and agree) with their reasoning concerning the validity of the spin polaron picture beyond the $\sim 5\%$ doping. That is why I would like to recommend this manuscript for publication in Nature Communications -- provided that also the paper abstract is slightly modified so that to include the fact that the presented hypothesis relies on the (expected) validity of the spin polaron picture in the underdoped cuprates.

Referees Report: NCOMMS-23-63796-T Grusdt et al

I've tried hard to understand this paper and found it to be very difficult as the authors are building on lots of self-referencing involving their rather large and complex body of work. The notation is not standard but rather borrows from the particle physics world, as they refer to "partons", "mesons", geometric string theory, etc. I do not believe this notation adds clarity and the paper would benefit by dropping it. Indeed, this paper needs a considerable amount of work to be useful to the readership and even then I am not sure it will reach that goal. In short, it devotes too much of the discussion to the technical details of the derivations and not enough to the big picture implications and experimental tests.

At the end of the day based on numerical solutions of a very dilute t-J model the authors seem to have arrived at a Feshbach picture in which the pairing follows the template of Feshbach-driven BCS-BEC in the atomic gases. This is a boson-fermion model in which the boson represents a tightly bound d-wave pair of holes in an antiferromagnetic host and the fermions involve combined "spinon-chargon" entities having a considerable sub-structure. It is these more complex fermions which are different from the analogues in atomic Fermi gases. This sub-structure in the fermionic channel also differentiates their "fermions" from other Feshbach models in the solid state such as Ranninger's resonating bi-polaron picture of the cuprates. There are, as well other such models being contemplated (see papers by Imamoglu) involving for example excitons in bilayer superconductors which can and should be discussed and compared.

It is certainly of interest to arrive at new ideas for pairing mechanisms particularly in cuprates and more strongly coupled superconductors. But to make this paper useful the authors need to emphasize the larger class of Feshbach models and, importantly, what experiments can be used to establish or falsify such a boson-fermion approach to superconducting pairing. Here, as the authors note, there are two types of pairs and this is true for entire class: the Feshbach boson (or two hole bound state in their rendition) along with the open channel fermion pairs. It would seem that this aspect could be tested experimentally?

Among the experiments they suggest (coincidence ARPES, coherent pair tunneling, pump probe experiments) which of these would be applicable to the entire Feshbach class? Which experiments are more specific tests of their physical picture? What about noise spectroscopy experiments?

I find it to be a major deficit that this Feshbach scenario does not have anything to say about the superconducting dome in the cuprates. It addresses only the extreme underdoped regime. Are there arguments for why T_c is actually lowest here and tends to increase with increased doping? The authors need to weigh in on a more extensive range of cuprate parameters.

In summary, this paper is not suitable for publication in its current form. The technical details which make up the bulk of the paper are far less interesting than their experimental tests and general discussion about related models for superconductivity. Without some major rewriting I do not see how this paper is appropriate for publication in a journal such as Nature Communications.

Response to the Reviewer #1

Reviewer:

I've tried hard to understand this paper and found it to be very difficult as the authors are building on lots of self-referencing involving their rather large and complex body of work. The notation is not standard but rather borrows from the particle physics world, as they refer to "partons", "mesons", geometric string theory, etc. I do not believe this notation adds clarity and the paper would benefit by dropping it. Indeed, this paper needs a considerable amount of work to be useful to the readership and even then I am not sure it will reach that goal. In short, it devotes too much of the discussion to the technical details of the derivations and not enough to the big picture implications and experimental tests.

Our reply:

We would like to thank the Reviewer for the feedback on how to improve our manuscript. However, we respectfully disagree that the notation used in our manuscript is "not standard". The parton description of charge carriers in underdoped cuprates traces back to early studies of high-Tc superconductivity. To our knowledge, string-like excitations were first described in *Trugman, PRB 37 (1988)* cited over 500 times in the field.

On the left, we reprint Figure 1 of the above Reference, which shows the geometric string object commonly used in the description of doped antiferromagnets.

First numerical evidence of string-like single hole spectra and their relation to mesons were found by *Beran et al., Nuc. Phys. B 473 (1996)*. Later, this string picture was more rigorously formulated by *Brunner et al., PRB 62 (2000)* and *Manousakis, PRB 75 (2008)*. A more quantitative, strong coupling description of the single hole in an antiferromagnet was developed by some authors of this manuscript, among numerous other researchers in the field.

Additionally, the tightly-bound charge $2e$ boson, i.e. the closed channel meson in our model, has likewise been studied in the literature before. We refer to *Poilblanc et al., PRB 49 (1994)*, in which the internal rotational structure of the charge $2e$ boson was observed in numerical studies.

We highlight that the string picture is fully consistent with ARPES spectra in the very low-doping regime and agrees with numerical simulations of the Fermi-Hubbard and t-J model – importantly it is equivalent to a strong-coupling description using the underlying electrons "c". However, the parton/meson picture (= bound state of a spinon and chargon) provides a useful perspective. For example, in *Punk et al., PNAS 112 (2015)*, a tightly-bound spinon-chargon "dimer" is introduced to construct a fractionalized Fermi liquid (FL*). These "dimers" have the same quantum numbers as our (sc) mesons, but we include the internal meson structure recently observed in ultracold atom experiments as well as in numerical studies, see Refs. [81,94].

Even more, it is the parton description that allows us to identify the constituents of the two-channel model, which we put forward in our work. In principle, the two types of state (sc + sc) and (cc) can be formulated in terms of the underlying $\hat{c}_{j,\sigma}$ operators, but are much too cumbersome to work with. For example, the parton formalism has shown to be useful in the 1D Hubbard model, where non-local string operators between spinons and chargons can be defined in terms of the underlying $\hat{c}_{j,\sigma}$ operators. While in the 1D system the Hubbard model exhibits spin-charge separation, the physics is very different in the 2D Fermi-Hubbard, where the spinons and chargons can be confined at low doping as described in our study.

In this vein, we believe to provide a novel big picture perspective on the underdoped cuprates: we develop a coherent framework in terms of meson-like constituents by including the numerous experimental, numerical and analytical evidences of the internal structure of charge carriers at low doping. Our proposed two-channel model offers a new explanation for the origin of strong pairing of charge carriers and we discuss several different probes for solid state experiments as well as cold atoms.

Reviewer:

At the end of the day based on numerical solutions of a very dilute t-J model the authors seem to have arrived at a Feshbach picture in which the pairing follows the template of Feshbach-driven BCS-BEC in the atomic gases. This is a boson-fermion model in which the boson represents a tightly bound d-wave pair of holes in an antiferromagnetic host and the fermions involve combined "spinon-charge" entities having a considerable substructure. It is these more complex fermions which are different from the analogues in atomic Fermi gases. This sub-structure in the fermionic channel also differentiates their "fermions" from other Feshbach models in the solid state such as Ranninger's resonating bi-polaron picture of the cuprates. There are, as well other such models being contemplated (see papers by Imamoglu) involving for example excitons in bilayer superconductors which can and should be discussed and compared.

Our reply:

We thank the reviewer for the summary and comparison of our work to previous studies. We fully agree that our model picks up on many interesting phenomena in the context of BEC-BCS crossover. As the referee points out, our model is distinct from previous work because of the microscopic structure we describe, specifically we include "more complex fermions". Indeed, we find that these complex fermions with their internal structure naturally explain the d-wave pairing interactions and it is their finite spatial extent, which leads to non-zero coupling matrix elements between the two channels.

Further, the earlier work by Atac Imamoglu's group and others was very much an inspiration for our work and we already cite the papers in the abstract of our manuscript. Those experiments demonstrate a mechanism to engineer strong interactions between exciton quasiparticles in solid state devices, which is a very interesting direction. But we believe that an extended discussion and comparison would go beyond the scope of our manuscript. We are aware that in the meantime new work in this direction has appeared, which we are now citing in the revised version of our manuscript as well – highlighting the timeliness of our approach.

Hence, our key message is not that we reinvent Feshbach-induced pairing but rather find a scenario how Feshbach resonances may play a significant role in doped antiferromagnetic Mott insulators/cuprates. This includes identifying the two-channel model in the first place, calculating the effective interaction for two particles in the open channel and elaborating on the consequences for underdoped cuprates - based on symmetries and selection rules.

Reviewer:

It is certainly of interest to arrive at new ideas for pairing mechanisms particularly in cuprates and more strongly coupled superconductors. But to make this paper useful the authors need to emphasize the larger class of Feshbach models and, importantly, what experiments can be used to establish or falsify such a boson-fermion approach to superconducting pairing. Here, as the authors note, there are two types of pairs and this is true for entire class: the Feshbach boson (or two hole bound state in their rendition) along with the open channel fermion pairs. It would seem that this aspect could be tested experimentally?

Our reply:

We thank the referee for acknowledging that our mechanism is an interesting new idea for pairing in cuprates or even a larger class of superconductors. We would like to refer to the "Perspectives" section in our manuscript discussing the relevance with respect to a broader class of superconductors.

While Feshbach-induced pairing has raised significant attention in the solid state context in the past years, see e.g. *Schwartz et al., Science 374 (2021)*; *Crépel and Fu, Sci. Adv. 7 (2021)*; *Yang et al., arxiv:2309.15095*, there is to our knowledge not *the* universal signature that would provide a "smoking gun" signature of a Feshbach scenario. Of course, signatures of attractive and repulsive branches constitute important indications of a possible resonance, but tuning through the resonance in strongly-correlated systems is rather model specific. For example in exciton semiconductors the electric field can be used to tune the resonance. As we discuss in our manuscript, extended Hubbard interactions may be a useful tuning knob for Hubbard-like models.

Thus, in our manuscript we focus on probing the Feshbach scenario by experiments tailored for cuprates/Hubbard models. Hereby, we would like to highlight that we already devote an entire section of our

manuscript on concrete experimental probes in cuprate compounds. We reiterate that the existence of the boson channel across the underdoped regime would be a smoking gun signature of the two-channel model we propose. Further, the existence of the boson channel provides a numerical test case that can in principle be considered in more detail by large-scale computational studies.

Additionally, we want to share new preliminary, numerical studies from our group that predict a striking signature attributed to the Feshbach scenario. Within the phenomenological, truncated basis approach that we use to describe the open and closed channel, we can calculate the closed channel dispersion and study the effect of couplings to the open channel using a self-consistent Born approximation. Interestingly, we find a characteristic splitting of the pair dispersion into two strongly coupled branches that is a unique hallmark of two near-resonantly coupled channels; in particular such splitting only appears in the presence of the open-closed channel coupling, see Figure below. This characteristic signature can indeed be found in the DMRG calculation of the t-J model in Ref. [54].

We show preliminary results for the two-hole ARPES spectrum in the presence (middle) and absence (right) of coupling to an open channel, using the open-closed channel coupling matrix elements derived in our manuscript and using self-consistent Born approximation. On the left we compare to DMRG data for the same parameters (t-J model) from Ref. [54]. We interpret the splitting of the (cc) branch for non-zero momenta as a direct signature of two channel physics in doped antiferromagnets.

Reviewer:

Among the experiments they suggest (coincidence ARPES, coherent pair tunneling, pump probe experiments) which of these would be applicable to the entire Feshbach class? Which experiments are more specific tests of their physical picture? What about noise spectroscopy experiments?

Our reply:

We emphasize that our work does not aim to review physics of Feshbach-mediated pairing in a solid state system, but rather we find evidence that such a scenario may be realized in cuprate superconductors. In our opinion, it is sufficient in this paper to focus on experimental probes relevant for cuprate compounds, as we have discussed above. We believe that any direct experimental evidence of the charge $2e$ boson (closed Feshbach channel) will provide new insight into the nature of underdoped cuprates and doped Mott insulators in general.

That said, while suggestive, the mere demonstration of the charge $2e$ boson will not be sufficient to confirm the entire Feshbach scenario we propose. In addition, one needs to confirm that the two channels are strongly coupled. Fortunately, direct evidence for such coupling is provided by the splitting of the pair spectrum into sub-branches featuring level repulsion — as described in our reply to the last question by the referee — for which we have found preliminary numerical evidence (see Figure above) that we will publish in a forthcoming work. Indeed, such ‘avoided level crossing’ in appropriate spectra can be used for the entire class of Feshbach systems and is routinely applied to confirm this physical picture. For example, in the semiconductor systems mentioned before the two repelling levels associated with the Feshbach resonance can be understood as manifesting in repulsive and attractive polaron branches.

Furthermore, in the underdoped cuprates indirect evidence for the existence of stable bound states of two charges connected by a string readily exists, namely in the context of Hubbard-Mott excitons. Since the

charge carriers are constituted by a doublon and a hole in that case, the situation is not identical to the two holes discussed in our work, but a close relation can be established, see *Bohrdt et al., arXiv:2406.16854*.

Reviewer:

I find it to be a major deficit that this Feshbach scenario does not have anything to say about the superconducting dome in the cuprates. It addresses only the extreme underdoped regime. Are there arguments for why T_c is actually lowest here and tends to increase with increased doping? The authors need to weigh in on a more extensive range of cuprate parameters.

In summary, this paper is not suitable for publication in its current form. The technical details which make up the bulk of the paper are far less interesting than their experimental tests and general discussion about related models for superconductivity. Without some major rewriting I do not see how this paper is appropriate for publication in a journal such as Nature Communications.

Our reply:

Without any doubt, it is an important step to establish a theory in the entire low-doping regime and study the pairing dome. This requires to build a model that adiabatically connects the distinct charge carriers in the underdoped and overdoped regime, see e.g. *Badoux et al., Nature 531 (2016)*. We are currently working on a "BCS-BCS" crossover type scenario, in which the Cooper pair branch on the overdoped side is adiabatically connected to the energetically higher lying tightly-bound $2e$ (cc) pair in the underdoped regime. As antiferromagnetism increases, the (sc) charge carriers emerge and form the ground state in the underdoped regime.

The described scenario is closely related to our discussion in the manuscript about the onset of superconductivity as doping is increased, see page 3. In the very low-doping regime and deep in the insulator, screening is expected to be less effective such that beyond on-site Hubbard interactions may not be neglected. These interactions play a significant role for the bosonic (cc) channel, such that the energy difference ΔE between the scattering channels increases at low doping and cannot overcome the repulsive background scattering of the fermionic holes.

That said, we agree with the referee that a central question remains what exactly is the physics of the superconducting dome, but we would also like to point out that no generally accepted theory exists to this day. Nevertheless our work provides a microscopic perspective that suggests the following scenarios:

- 1) A natural question is whether the near-resonant (Feshbach-type) interactions we describe at low doping may become resonant at or around optimal doping. This would lend a natural explanation for the existence of the superconducting dome, with critical T_c 's that can be expected to be some 20% of the Fermi energy E_F . In our manuscript we refer to this possible phenomenology by alluding to a significant body of previous work on BEC-to-BCS crossover physics that may be at play. The key new insight contributed by our current manuscript is that we provide a microscopically justified, strong-coupling picture how near-resonant interactions can indeed arise in cuprates. Indeed, on general grounds one expects unitary interactions approximately when the binding energy ΔE of the closed channel (cc) bound state is comparable to E_F . Since E_F is strongly doping dependent, a sharp rise in T_c can be expected at low doping until the optimal T_c is reached. The known orders of magnitude of Fermi energies and binding energies are entirely consistent with having unitary interactions at optimal doping.
- 2) If a scenario involving unitary interactions and a BEC-to-BCS crossover actually applies, one needs to wonder where the BEC phase of tightly bound pairs is realized. In the BEC-to-BCS literature in the context of cuprates, one typically assumes a BEC-type superfluid on the underdoped side. This picture appears to be inconsistent however with the phenomenology of the Fermi-arcs which in many ways resemble the free (sc)-type magnetic polarons that we use as a starting point (see e.g. *Sous, He and Kivelson, npj Quantum Materials 8:25 (2023)* for a recent discussion). Since our theoretical model provides a microscopic explanation how (near-) resonant interactions can naturally arise among unpaired fermionic constituents, we solved a major problem of the phenomenological BEC-to-BCS

description of cuprates. At even higher dopings beyond optimal doping, we are currently exploring the BCS-to-BCS crossover scenario alluded to earlier.

- 3) The rise of T_c with increasing doping is generally expected for models in which the density of charge carriers increases with doping, see e.g. *Uemura et al., PRL 62 (1989)*. This perspective is consistent with our description: We start from the quasiparticles of doped holes in an antiferromagnetic Mott insulator. As we argue, the magnetic polarons form a weakly interacting Fermi sea and hence the doping level corresponds to the charge carrier density, i.e. the density of magnetic polarons or doped holes, establishing the onset of superconductivity at very low doping.

We are of course very excited about scenarios 1) and 2) above, which would indeed provide a significant new perspective — based completely on microscopic insights to the doped Hubbard-type models — on the superconducting dome observed at higher dopings. Developing full understanding of the implications of the Feshbach pairing model to the superconducting dome would be equivalent to providing a full solution of the high- T_c problem. While it is certainly something that we are working on, we believe that the current paper provides more than enough new ideas and results to justify a separate paper.

In summary, the parton picture in the underdoped cuprates is a standard terminology in the field and, in our case, allows us to identify the constituents of the two-channels. In our manuscript, we formally introduce the two-channel formalism, which is necessary to understand the symmetry arguments underlying the selection rules we derive. Ultimately, this allows us to derive a microscopic model of how strong d-wave pairing interactions can arise in doped antiferromagnets consistent with experiments on cuprates. Further, we discuss four experimental probes to test our hypothesis and briefly discuss probes for cold atoms and for numerical studies. We emphasize that our findings are relevant for cuprate superconductors, for which we can provide strong arguments. Of course, we agree that it is of great interest to study the Feshbach scenario for other families of strongly-correlated superconductors. One example is constituted by bilayer nickelates under pressure, for which our group has recently proposed the same Feshbach scenario to underly superconductivity, see *Lange et al., PRB 109 (2024)*; *Lange et al., arXiv:2309.13040*; *Schloemer et al., arXiv:2311.03349*; and also *Yang et al., arXiv:2309.15095*. However, we believe a further detailed discussion of other systems is beyond the scope of the current work and overall would be too speculative at the moment. In light of the relevance and potential implications of our proposed pairing mechanism, we hope that the reviewer can recommend our revised manuscript for publication in *Nature Communications*.

Response to the Reviewer #2

Reviewer:

The manuscript by L. Homeier et al. touches on the old but still largely unresolved problem, i.e. on the origin of superconductivity in 2D Hubbard-like models. To this end, it uses a speculative hypothesis which postulates that a Feshbach resonance between the fermionic mesons and the bosonic two-hole states may explain the apparent pairing in this class of models. While certainly such studies are needed, and the concept in principle is interesting, I doubt whether this study can be published in Nature Communications. In general, I find the study, even for a "hypothesis-like" paper, way too speculative, since a number of reasoning elements are either entirely skipped or are not really clear. More precisely, I have the following issues with the presented study:

Our reply:

We would like to thank the Reviewer for the evaluation of our manuscript and to recognize the significance of the "unresolved problem" we consider in our study. We believe that we can address all points raised by the referee in our detailed reply below.

Reviewer:

(I) Validity of the "sc" description:

(a) It is written that the "sc" fermionic meson is *the* quasiparticle that is observed in e.g. ARPES on cuprates or in the spectral function of the Hubbard / t-J models. However, such an "sc" bound state binds a hole (c) with a "neighbouring" spin (s) -- and in ARPES the quasiparticle response is actually widely-understood (cf. [69, 71, 95, 134]) as originating just from a single hole. Therefore, the Authors' approach goes against the current understanding and needs further (and substantiated) explanation.

Our reply:

We fully agree with the reviewer that the one-hole ARPES spectrum originates entirely from a single hole. The parton picture, in which we describe the hole as a bound state of a spinon and a chargin, also fully agrees with this picture. This is because the spinon carries spin but no charge, whereas the chargin carries charge but no spin, so a confined pair of a spinon and a chargin together form an electronic entity -- the doped hole, or magnetic polaron.

Let us explain the ARPES spectrum in the parton picture: Both the bare electron as well as the "sc" bound state carry the quantum numbers spin (s) and charge (c). However, the "sc" bound state does not necessarily describe a point-like particle but instead it is a spatially extended object with finite probability of finding the spinon and chargin at the same site. Thus, the spectral weight $Z(k, \omega)$ measured in ARPES has an intuitive interpretation in the parton picture: it corresponds to the probability $p(\ell = 0)$ of the string length $\ell = 0$ state at given momentum and energy. We note that one misconception may have been that the (s) and (c) constituents bind on neighbouring sites: although that is a very likely configuration we do include any length $\ell \geq 0$ with most weight located at very short strings, see *Grusdt et al., PRX 8 (2018)*.

Therefore, the parton picture and the picture of the underlying bare electrons are *equivalent* and constitute alternative descriptions of one and the same problem. Both pictures are commonly used in the theoretical description of the single-hole problem, see Refs. [66-79], and to our knowledge were first studied in the context of high-Tc by *Béran, Poilblanc and Laughlin, Nuc. Phys. B 473 (1996)*. Also later extended parton objects have likewise been studied, e.g., in *Punk, Allais and Sachdev, PNAS 112 (2015)* they consider a dimer object with partons on neighbouring sites. For a thorough discussion we refer to *Bohrdt et al., PRB 102 (2020)*.

Reviewer:

(b) What are the explicit conditions that (3) is valid? What is the explicit equation for the dispersion relation in (3) and how it is obtained from the Hamiltonian (2)?

Our reply:

In Eq. (3) of our manuscript, we model the (sc) fermions as a Fermi liquid with a dispersion relation (see below) that gives rise to a Fermi surface resembling the hole pockets. At first, this is a commonly accepted approach, which captures the established facts in the underdoped cuprates:

- (i) The volume of the Fermi surface scales with the hole doping δ up to about 19%, where the nature of charge carriers suddenly changes, see e.g. *Doiron-Leyraud et al., Nature 447 (2007)*, or *Badoux et al., Nature 531 (2016)*.

On the left, we reprint Figure 4b of *Badoux et al., Nature 531 (2016)* showing the abrupt change in Fermi surface volume obtained from Hall coefficient measurements. The open channel description Eq. (3) in our manuscript models pockets of magnetic polarons (holes) with a Fermi surfaces centered around $k = (\pm\pi/2, \pm\pi/2)$ with volume $\propto \delta$. It is one of the open questions in underdoped cuprates, when and how exactly the hole pockets break down.

- (ii) As doping is increased, the Fermi surface gets modified from hole pockets to Fermi arcs to a large Fermi surface, see *Shen et al., Science 307 (2005)*. At very low doping, recent laser ARPES measurements of the hole pockets are consistent with the picture of a Fermi liquid of magnetic polaron constituents, see *Kurokawa et al., Nat. Commun. 14 (2023)*. In the low-doping regime, mechanisms for the pseudogap and the Fermi arcs have been proposed by Sachdev and collaborators. In these proposals, the Fermi liquid is coupled to a topological state (FL*). Likewise, we could adapt our microscopic ansatz, Eq. (3), into the pseudogap regime by coupling the (sc) fermions to a topological phase.

These observations confine the regime, in which the (sc) picture may be valid in cuprate superconductors. The experimental observation of polaron-type signatures consistent with the string picture up to about optimal doping in quantum gas microscopes further corroborate this picture, see *Koepsell et al., Nature 572 (2019)* and *Koepsell et al., Science 374 (2021)*.

Unfortunately, there is no exact analytical way to get the (sc) dispersion from the t-J model at finite doping. Therefore, in our manuscript we use a heuristic approach to obtain Eq. (3) from the t-J model in Eq. (2) by elevating the single-hole problem to the finite doping regime. We refer to concerted numerical and analytical efforts over the past decades that have obtained the single-hole dispersion, see Refs. [66-79, 134]. Here, we use the dispersion relation derived in *Martinez et al., PRB 44 (1991)* given by

$$\varepsilon_{sc}(k_x, k_y) = A[\cos(2k_x) + \cos(2k_y)] + B[\cos(k_x + k_y) + \cos(k_x - k_y)],$$

where A, B are fit parameters that depend on mostly on J (and only weakly on t) when $t \gg J$. This dispersion relation is consistent with ARPES spectra of the single-hole in cuprates, see e.g. *Wells et al., PRL 74 (1995)*. Simple analytical estimates for A and B were derived, e.g., in *Grusdt et al., PRB 99 (2019)*. In Eq. (3), we assume that the (sc) fermions fill up a Fermi sea according to the above dispersion relation. We note that thus far we can only provide a quantitative description in the very low-doping regime, see also *Bermes et al., PRB 109 (2024)*.

We fully agree that it is useful for the reader to provide the above expression for the (sc) dispersion relation, which we now explicitly printed in Eq. (9) in the revised Methods section.

In general, the above discussed approach describes the philosophy of our Feshbach hypothesis: (i) In the very low-doping regime, we can explicitly calculate properties of the two-channel model based on semi-analytical descriptions of open and closed channel states. We find evidence for a Feshbach resonance, and

show that the resulting pairing mechanism has d-wave symmetry. (ii) There is empirical evidence that the magnetic polaron picture remains valid in the underdoped regime. (iii) We hypothesize that the Feshbach mechanism is valid throughout the underdoped cuprates. We leave it to future studies whether the Fermi liquid description of magnetic polarons remains a good description or whether a more exotic FL* parent state has to be constructed. Regardless of the microscopic details of the underlying parent state, we conjecture that the Feshbach scenario adiabatically connects the very low-doping and optimally doped regimes.

Reviewer:

(II) Validity of the "cc" description:

(a) Papers [54] and [88] (that form the basis of the "cc" dynamics description) seem to contradict each other: according to [54] the tightly-bound pair disperses by $\sim t$ but according to [88] the tightly-bound pair dispersed by J_{\perp} . Which one is correct and why?

(b) In the current manuscript it is assumed that "cc" moves due to J_{\perp} and paper [54] is cited there -- but, as just written above, paper [54] claims that "we find a long-lived, tightly bound state of two holes, which can move as fast as the hole hopping t ." (page 3 of [54]).

Our reply:

We thank the reviewer for the detailed reading. References [54] and [88] have revealed two types of (cc) states co-existing in the spectrum: Depending on the C4 symmetry of the "cc" pair, the dispersion is either $\propto t$ (s-wave) or $\propto J_{\perp}$ (d-wave). Let us summarize and compare Refs. [54] and [88].

Ref. [88]: In this study, the geometric string formalism is applied to the case of two tightly-bound holes. The string picture is most accurate in the t - J_z model underlying this study, i.e., $J_{\perp} = 0$ was assumed there. It was found that the s-wave pair disperses strongly $\propto t$, whereas the dispersion of the d-wave pair is exactly flat, which can be understood from destructive interference.

Ref. [54]: Here, the pair spectrum is obtained from time dependent matrix product state techniques of the t-J model, i.e., quantum fluctuations are included and $J_{\perp} = J_z = J$. While this confirms the strongly dispersive ($\propto t$) s-wave "cc" pair, it find that the d-wave pair disperses with J_{\perp} attributed to magnetic fluctuations. In addition, it shows that the spectral lines are strongly visible above the ground state, which is interpreted as long-lived excited charge $2e$ pairs consistent with the string-like parton bound states. Similar phenomenology was recently found for Hubbard-Mott excitons, see *Bohrdt et al., arXiv:2406.16854*, in agreement with experiments on cuprates.

Hence, the two studies are fully consistent when sending $J_{\perp} \rightarrow 0$ in [54]. Ref. [54] takes the full t-J model into account revealing additional features of the "cc" dispersion. The energetically low-lying "cc" state has d-wave symmetry, see also *Poilblanc et al. PRB 49 (1994)*, and we argue that this pair constitutes the relevant scattering channel in the Feshbach hypothesis, because the scattering length scales with the inverse energy difference $\Delta E_{m_4}^{-1}$ between the open- and closed channel, which is the largest for the d-wave (cc) state. This reduces our model to a two-channel model and we only consider the d-wave pair, which disperses with J_{\perp} . We refer to our discussion on multichannel scattering models in *Homeier et al., PRB 109 (2024)*.

Reviewer:

(c) I could not find the explicit equation for ε_{cc} in [54] so I would appreciate if the Authors explain ("step-by-step") how it is obtained from the Hamiltonian (2).

Our reply:

We thank the reviewer for the comment and for their careful reading of our manuscript. Indeed, we have missed a factor of $2/3$ in the dispersion relation, which we have corrected in the revised version of our manuscript. This factor does not change any conclusions of our work.

We are happy to give a step-by-step explanation on how to obtain the chargon-chargeon dispersion $\varepsilon_{cc}(Q)$. The explicit expression is not given in Ref. [54] but we can read-off the dispersion relation from the fitted 1D cuts:

- (i) in the caption of Figure 4, Ref. [54], the numerically obtained dispersion of the d-wave pair is well described by the function $\varepsilon_{cc}(Q_x, 0) = -\frac{2}{3}J \cos(Q_x) + b$, where b is a constant.
- (ii) From the C4 symmetry of the underlying t-J Hamiltonian, we conclude that the 1D cut along the Q_y -direction must have the same form $\varepsilon_{cc}(0, Q_y) = -\frac{2}{3}J \cos(Q_y) + b$
- (iii) Next, we consider individual points of the visible part of the dispersive band, e.g., $Q = (\pi/2, \pi/2)$ with $\varepsilon_{cc}(\pi/2, \pi/2) = b$. Note that the spectral weight vanishes at $Q = (\pi, \pi)$.
- (iv) This leads us to our ansatz $\varepsilon_{cc}(Q_x, Q_y) = -\frac{2}{3}J [\cos(Q_x) + \cos(Q_y) - 2] + \Delta E_{m_4}$. As explained in the manuscript, it is challenging to determine the absolute energies accurate enough with the current numerical resolution. Thus, the strategy is to use the functional form of the dispersion and introduce the phenomenological energy offset ΔE_{m_4} , which measures the relative energy between "cc" pairs with center-of-mass momentum $Q = 0$ and the Fermi surface of the open channel fermions. From empirical observations, see above, we conjecture that $\Delta E_{m_4} > 0$ leading to a fermionic ground state in the two-channel model. Ideally, we would like to derive $\varepsilon_{cc}(Q)$ analytically, but that is a highly non-trivial task and hence we resort to numerics as described above.

Reviewer:

(III) The resonance and the recombination process:

(a) The cartoon of Fig. 2(a), which shows the scattering channel energies as functions of the distance between the constituents, should be backed by calculations.

Our reply:

We thank the reviewer for this comment, which we appreciate. However we emphasize that this cartoon is really a cartoon intended to highlight analogies with Feshbach resonances observed in, e.g., ultracold atoms or semiconductors and parts of it cannot be taken too literally: First of all, we consider a system with strong lattice effects, so distances are discrete; More importantly, for the closed channel states (cc) we consider a Hilbert space spanned by string states which cannot be labeled simply by distances. Instead, for each given string length (the closest analog to distance), there are exponentially many distinct string states with slightly different energies and tunnel couplings between them. For a pair of two (sc) states, we can define a potential as a function of their center-of-mass distance, but it is not easy to calculate, involves overlaps of magnonic dressing clouds, and most importantly is not believed to play any role for our discussion besides influencing quantitative values for scattering length that anyway depends on our fit parameter ΔE .

Some elements of the cartoon in Figure 2a are accurate, and relevant for the calculation of the scattering length we perform: Namely, the existence of discrete bound states in the closed channel (horizontal lines), with an energy gap $\mathcal{O}(t)$ between d- and s-wave states; the qualitative linear rise of the indicated effective (cc) potential with distance (indicating confinement); the position of the d-wave state slightly above the scattering threshold.

If the referee feels strongly that the scattering potential is misleading, which does not play any important role for our discussion, we are happy to remove this from the figure before publication.

Reviewer:

(b) The physical mechanisms behind the recombination process are not really clear / consistent:

-- On one hand, t' is not included in the description of "cc" and "sc" states (including t' would qualitatively affect their dispersion relations). Justification of this approximation should be given.

-- On the other hand, J_{\perp} is already crucial for the mobility of "cc" states so it is not clear to me how it can at the same time lead to the recombination process shown in Fig. 2(b). This double role of J_{\perp} should be explained.

Our reply:

The referee addresses an important point in our two-channel model. How do the terms in the Hamiltonian couple between and within the channels?

As shown in the sketch on the left, all terms in Hamiltonian Eq. (2) will affect the properties of the open and closed channel dispersions. Additionally, the terms $\propto t', J_{\perp}$ introduce couplings between the open and closed channel. We believe this should clarify the "double role" of the couplings J_{\perp}, t' : they lead to matrix elements between state within open and closed channel states, as well as between states from different channels.

Regarding the role of t' for the dispersion, the referee is absolutely correct that we have to include t' in the dispersion relations if we are interested in quantitative calculations. However, for sizable t' , but $t' < t, J$ the qualitative symmetry properties of the open and closed channel do not change, such that the resulting pairing interaction will remain in the $d_{x^2-y^2}$ channel. Moreover, it is an experimental fact in the cuprates that hole pockets form around $k = (\pi/2, \pi/2)$ that we capture in our effective two-channel description. This supports the robustness of our proposed Feshbach pairing mechanism with respect to material parameters.

We have dedicated the discussion and role of t' to an accompanying paper by some of the authors, see *Homeier et al., PRB 109 (2024)*. In this paper, we construct the analytical framework of the two-channel model based on geometric strings. In particular, we apply a refined truncated basis method, which basically takes into account loop effects in a more systematic and numerical way. This numerical approach allows us to include t' in the calculation of the meson wavefunction and open-closed channel recombination process. Moreover, it shows that – once the formalism is established – it is rather straight forward to study effects of t' or other terms in the Hamiltonian.

The Figure below compares the "simple" geometric string theory to the refined truncated basis method. The latter fully includes $t'/t = -0.2$ in the dispersion relation. As argued above, the d-wave pairing symmetry of the form factors $M_2(k)$ remains robust. Nevertheless, we find a quantitative change in the position of the additional "nodal ring" moving towards the center of the Brillouin zone. Hence, to understand quantitative features of the Feshbach pairing mechanism one has to carefully take into account all terms of the Hamiltonian at all steps of the calculation. In future studies, one could take into account even more terms, such as the typically ignored 3-site term in the t-J model, or density-assisted tunneling terms that arise from downfolding methods, see e.g. *Jiang et al., PRB 108 (2023)*.

Once again, we are grateful for the referee's constructive comments and excellent, detailed questions; we have adapted and improved our manuscript accordingly. In our above reply, we could address all of the referee's points and hope to have convincingly demonstrated the "hypothesis-like" approach of our manuscript: The Feshbach mechanism is an original and novel proposal for the origin of strong pairing in cuprates, and further (i) it is consistent with empirical observations in cuprates, (ii) it can be extended to other parent states, such as FL* and (iii) we propose concrete experimental and numerical probes to search for the two-channel physics in cuprates.

We hope the Reviewer can now recommend our manuscript for publication in *Nature Communications*.

Top: We show the scattering form factor for spin-flip J_{\perp} (left) and NNN tunneling processes t' (right) obtained from the geometric string model, where the form of the variational bound state wavefunction only accounts for t and J . Bottom: A refined truncated basis method, which also carefully treats the overcompleteness due to Trugman loop, see *Homeier et al., PRB 109 (2024)*, allows us to include the non-perturbative effects of t' to the meson wavefunctions. While important quantitative features are affected by the different treatments, qualitative features, such as the $d_{x^2-y^2}$ symmetry or the overall scale of the scattering form factors, agree in both descriptions.

Response to the Reviewer #3

Reviewer:

This paper advances a theory of pairing in strongly correlated fermionic Hubbard-type models. The emergent spinon and chargeon entities, which respectively carry spin and charge and are fermionic, can bind into bosonic spinon-chargeon (sc) and chargeon-chargeon (cc) "mesons." The sc mesons undergo strong scattering and the Coulomb repulsion between them is apparently overcome by a Feshbach resonance between these emergent excitations.

This paper is very clear and well-written. The figures are all nicely done and the sketches are helpfully evocative. As it is not possible to show from first principles whether a given Hubbard-type model will result in the Feshbach resonance pairing physics for the notional emergent partons, the authors appropriately focus on describing experimental signatures of their scenario suggest experiments based on coherent tunneling of Cooper pairs as well as predicted signatures within ARPES.

Our reply:

We thank the reviewer for the nice summary of our manuscript. Moreover, we are pleased and happy about the Reviewer's comment on the presentation of our manuscript.

Reviewer:

Fundamental aspects of the Feshbach resonance pairing mechanism are also articulated in ref. 61, which describes a theory of pairing via Feshbach resonance in nickelate superconductors. However, there is much that is new in the current submission, which is specifically relevant to cuprates. There is only a faint relation to the work in Rocchina Caivano et al. (Supercond. Sci. Technol. 22, 014004 (2009)) and in the paper of Squire and March (Int. Jour. Quantum Chem. 110, 2808 (2010)).

Our reply:

The referee puts our work in context to similar and previous literature, and acknowledges the novelty of our work. We want to emphasize that we do not want to make the impression of "re-inventing" Feshbach resonances. Especially in the past years we recognize increasing attention in Feshbach-induced pairing interactions, see e.g. *Schwartz et al., Science 374 (2021)*; *Crépel and Fu, Sci. Adv. 7 (2021)*; *Yang et al., arxiv:2309.15095*.

The main result in our work is to identify the constituents of a two-channel model in cuprate compounds. By analyzing their symmetry properties, we arrive at a set of selection rules that we combine with numerical and empirical insights to formulate our Feshbach hypothesis for underdoped cuprates.

Further, we thank the reviewer for highlight two very interesting references, which we are now citing when we discuss Feshbach pairing in related systems.

Reviewer:

Overall, the theory and suggested experimental probes are speculative. It is not clear to me whether any of the experimental signatures proposed could not also be present in a model which did not invoke the Feshbach resonance.

The theoretical work here is very strong indeed, but would be more compelling if accompanied by comparison with actual experimental data.

Our reply:

We fully agree with the reviewer that actual experimental data is needed to verify our hypothesis. The goal of this paper is to discuss several experimental and numerical probes that are suitable to identify a low-lying closed channel. Moreover, we believe that tuning parameters such as extended Hubbard interactions may be suitable for modifying the energy splitting ΔE_{m_4} between open and closed channel, such that resonance features could be probed numerically or in quantum simulation. Experimental progress in either of the suggested experiments will constitute a key step toward establishing the Feshbach pairing hypothesis.

The reviewer raises the important question which signatures can single out the Feshbach hypothesis from alternative models. We believe two steps need to be taken: (1) in addition to the established 1-hole (sc) channel (that has been directly observed in ARPES on cuprates), the existence of the closed channel needs to be proven (see above); In addition, (2), experiments need to demonstrate strong coupling, i.e., hybridization, between the two channels: this is the hallmark of the Feshbach hypothesis, which would be very hard to explain by any other theories. In our manuscript we focused on the first experimental step, i.e., the observation of the closed channel. However, we would like to take this chance to share with the reviewer an exciting new, preliminary numerical result addressing directly the second point: The truncated basis method described in the manuscript provides a phenomenological description of the two-channel model. Within this analytical model, we can calculate the two-particle ARPES spectrum (revealing the closed channel dispersion) and study the effect of couplings to the open channel using a self-consistent Born approximation. Interestingly, we find a characteristic signature in the shape of the pair dispersion that is a unique hallmark of two-channel physics, i.e., which only appears in the presence of the open-closed channel coupling. This characteristic signature can indeed be found in the DMRG calculation of the t-J model in Ref. [54]. Therefore, we may have found a direct numerical signature that invokes the conjectured Feshbach resonance and that future experiments can search for. We print a preliminary figure with numerical results below.

We show preliminary results for the two-hole ARPES spectrum in the presence (middle) and absence (right) of coupling to an open channel, using the open-closed channel coupling matrix elements derived in our manuscript and using self-consistent Born approximation. On the left we compare to DMRG data for the same parameters (t-J model) from Ref. [54]. We interpret the splitting of the (cc) branch for non-zero momenta as a direct signature of two channel physics in doped antiferromagnets.

We would like to thank the reviewer again for the positive evaluation of our manuscript. Despite being preliminary unpublished results, we believe to have provided an idea of signatures that are unique to our proposed Feshbach scenario, and that can be tested in future experiments. This provides an exciting new perspectives for probes in cuprate superconductors and possible implications for the theoretical understanding of doped antiferromagnetic Mott insulators. We hope the Reviewer can recommend our manuscript for publication in *Nature Communications*.

List of changes to our manuscript:

- Fixing a prefactor in the (cc) dispersion relation after Eq. (4) [pointed out by Reviewer #2]
- Stating the dispersion relation of the (sc) mesons in the Methods [pointed out by Reviewer #2]
- Updating References

All modifications are further highlighted in the attached document "NCOMMS-23-63796-T_diff.pdf".

List of changes to our manuscript:

- Updated references
- Added a note about the relation between the parton picture and the conventional picture of single holes (inspired by Reviewer #2)
- Clarified the illustration of potential curves in Figure 2 (suggested by Reviewer #2)
- Added a paragraph explaining the equivalence between the parton picture and a description using the underlying electrons (inspired by Reviewer #1 and #2)
- Explained the origin of the dispersion relation Eq. (3) (suggested by Reviewer #2)
- Added a separate paragraph on Feshbach resonances in solid state materials (suggested by Reviewer #1)
- Provided a brief summary statement in the perspectives section
- Explained the necessary requirements for extending our model to finite doping in the main text and in the perspectives section (inspired by Reviewer #2)
- Added a new section in the Methods giving a pedagogical overview of the parton picture in doped antiferromagnets (inspired by Reviewer #1)
- Explaining the relation of the spectral weight obtained from single hole ARPES in the geometric string formalism in the Methods section (suggested by Reviewer #2)
- Added a paragraph and new figure in the Methods section illustrating the double role of couplings in the Hamiltonian (inspired by Reviewer #2)

All modifications are further highlighted in the attached document "NCOMMS-23-63796-T_diff.pdf".

Response to the Reviewer #1

Reviewer:

What I wanted in a revised manuscript was not delivered. My request of the authors was to make a greater effort towards pedagogy and to provide a more useful paper for the readership-- not just in an afterthought fashion as a two paragraph "Perspectives" section, but in greater depth. For example, I do not believe many experimentalists, even those working in high temperature superconductivity, will be able to take away a simple physical picture or be inspired to do future experiments.

Our reply:

We are grateful for the Reviewer's feedback on the pedagogy of our manuscript. In the revised version of our manuscript, we have now included extended paragraphs in the main text and a new section in the Methods that provides the reader with additional explanations on the parton picture we use in our study. Particularly, we explain that the parton picture is an alternative framework, which is fully equivalent to a description using the underlying $\hat{c}_{j,\sigma}$ electrons. We further provide a short derivation showing that single-hole ARPES can be consistently formulated in the geometric string language; we believe this helps to explain our physical picture to an even broader readership and address experimentalists in the field.

Reviewer:

I realize that this Journal aims to publish "important advances of significance to specialists", but in my view the specialists here are not just theorists but experimentalists as well. And this is an exclusively theoretical and, indeed speculative paper and would have been much stronger had there been some degree of experimental support. Predictions for future experiments are not sufficient to address this concern.

Our reply:

Experimental evidence will be crucial to support our Feshbach hypothesis in the underdoped cuprates and we fully agree with the Reviewer on this position. We would like to emphasize again that an entire section of our manuscript is dedicated to experimental signatures, including the discussion of four different probes in solids as well as an outlook for ultracold atoms. Due to the complexity of the Fermi-Hubbard model, the prediction of precise amplitudes for coincidence ARPES at finite doping is out of reach for any numerical method at the moment. In our view, the discussion of a variety of potential experimental or numerical probes provides a clear path towards an experimental confirmation of our prediction, making our manuscript of interest to the broad readership of *Nature Communications*.

Reviewer:

On the positive side this paper is ambitious and in this regard I applaud the authors. In principle, it can set a new direction for theory-- particularly if there is more than numerical support to follow. And particularly, it looks to have a broader relevance, going beyond a focus on the cuprates but addressing other correlated superconductors where a generic Hubbard model appears relevant. Lacking this support at this time, I think it might be more suitable for a Journal such as Physical Review X.

For these reasons I will not recommend its publication in Nature Communications.

Our reply:

We are thrilled by the kind words and appreciation of our theoretical work by the Reviewer. The broader relevance for models with strong repulsive interactions is now also more clearly addressed in the new section of our revised manuscript.

Our study is based on solid theoretical and numerical evidence at low doping of cuprates – any effort to provide further experimental or numerical verification of our hypothesis at finite doping would have large impact on the field of strongly correlated electrons and doped insulators. Our work delivers important advances of significance to specialists within the field of theoretical condensed matter physics, solid state experimentalists and the cold atom community. Hence, we believe our work deserves an outlet with high-impact and broad readership and, based on this consideration, we hope the reviewer will support publication of our manuscript in *Nature Communications*.

Response to the Reviewer #2

Reviewer:

First of all, I would like to thank the Authors for their detailed response to all my comments.

In general, I am quite satisfied to their replies to my comments I(a), II, and III: these replies have helped me much better understand the content of the paper and they have clarified some of my doubts. On these points, I just have one request: I would appreciate, if the replies to my concerns were more thoroughly and explicitly incorporated in the manuscript [in particular, Fig. 2(a) can stay as is but an explicit discussions of its shortcomings is needed, in my opinion].

Our reply:

We thank the reviewer for carefully considering our reply and we are delighted to have elucidated the previous doubts. To include parts of this reply in the revised version of our manuscript is an excellent suggestion and improves the clarity of our manuscript. We have modified our revised manuscript accordingly; in particular we have clarified the shortcomings of Figure 2(a). In addition, we now emphasize even more the explanations on the comments from the previous and current reply: (i) We explain the origin of the dispersion relation in Eq. (3); (ii) We elaborate on the relation between the parton picture and a description using the underlying “c” electrons, also in relation to the interpretation of ARPES measurements and its spectral weight; (iii) We discuss the extension of our Feshbach mechanisms beyond 5% doping, which is supported by recent numerical studies and cold-atom experiments; (iv) We dedicate a new section to Feshbach scenarios in other solid-state settings; (v) We provide a pedagogical overview of the parton picture in the Methods section; (vi) We add a section on the role of next-nearest neighbor tunneling and spin flip-flop processes in the Hilbert space of the two-channel model in the Methods section.

Reviewer:

However, reply to point I(b) makes me think that the whole discussion is only valid in an extreme regime of one / two holes -- which cannot be extended to the interesting (for the understanding of superconductivity in the cuprates or in the Hubbard model) regime of $> \sim 5\%$ doping. This is because the Authors assume that the “sc” bound states resemble the spin polarons, as known from the single-hole study. In fact, this assumption is central both to the form of Eq. (3) as well as to the derivation of the coupling between open and closed channel that follows from [Homeier et al., PRB 109, 125135 (2024)]. The latter statement is also clearly visible from cartoon in Fig. 2(b) in which the recombination process efficiently works, due to the spinon sitting not on the same site as the holon but rather next to it. On the other hand, already at *few percent doping* the dominant part of the ARPES spectrum of the Hubbard model is dominated by a “hole-like” quasiparticle and the spin polaron is strongly suppressed, cf. Fig. 1(b) of [Kohno, PRL 108, 076401 (2012)] or Fig. 3(b) of [Wang et al., Comm. Phys. 3, 210 (2020)]. So, in my opinion, the main question remains: Can the postulated mechanism (in particular, the coupling between closed and open channels) be at all valid once “ $l=0$ ” in the “sc” states?

Our reply:

The referee is addressing the most important and at the same time least understood regime, namely the underdoped cuprates but beyond 5% doping. Ultimately, the impact of our proposed Feshbach mechanism in the field will be decided by the validity of our approach in this region. While we cannot provide a rigorous proof of the Feshbach mechanism as the origin of strong pairing in the underdoped cuprates, we would like to put forward arguments and elaborate on promising indications from ultracold atoms experiments and numerical studies that suggest an underlying polaron picture up to dopings beyond 5%, possibly even optimal doping:

First, the magnetic polaron picture of the “sc” and “cc” bound states does not necessitate long-range antiferromagnetic order. Instead, short-ranged antiferromagnetic correlations of a few lattice sites, which are believed to be present across the entire underdoped regime, are sufficient to obtain an attractive string-like potential between spinons and chargons. In the presence of such short-range antiferromagnetic correlations, the recombination processes illustrated in Fig. 2(b) remain valid since they are based on local

terms in the Hamiltonian. Whether the normal state in the underdoped regime is well described by the magnetic polarons or whether there is a more complex underlying structure, such as in fractionalized Fermi liquid FL*, where (sc) acquire an emergent gauge charge, is therefore not important to our mechanism. Rather the local spin-hole correlations are the necessary ingredients for the Feshbach formalism. The starting point of the spin polaron at very low doping should be seen as a controlled and well-understood limit of the Feshbach mechanism. In the underdoped regime, the description will become more subtle but our picture should remain valid.

Second, this is further corroborated by ultracold atom experiments in optical lattices, which simulate the Fermi-Hubbard model. In Koepsell et al., *Science* 374 (2021) and Chiu et al., *Science* 365 (2019), the local string-like spin-hole correlations (or $\ell \neq 0$) were observed up to around 20% hole doping and at temperatures well above the critical T_c . More refined but not yet published data are consistent with string-like correlations in even higher-order correlation functions, see e.g. <https://meetings.aps.org/Meeting/MAR24/Session/Y30.5>.

Recently, two numerical studies, see Shackleton et al., arxiv:2408.02190 and Müller et al., arxiv:2408.01492, argue that such polaronic correlations are also consistent with an underlying FL* parent state; hence consistent with the assumption in our manuscript, namely the spinon and chargon not sitting on the same site.

Third, we thank the reviewer for raising our attention to these two nice studies on the spectral features of doped Mott insulators based on cluster perturbation theory. As the system is doped away from the single-hole case, we fully agree that the spectral weight will shift and additional features away from the Fermi surface appear in the spectrum. Predicting spectral features from a phenomenological model at finite doping is notoriously difficult. Nevertheless, we would like to point out that the magnetic polaron's dispersion relation is in good agreement with the shape of the hole pockets at low doping – not explaining the Fermi arcs, which suggests a more complex normal state such as the FL*. Indeed, a recent cellular DMFT study of ARPES spectra, see Bacq-Labreuil et al., arxiv:2312.14381, found signatures of magnetic (or spin) polarons up to at least 10% doping. Likewise, the evolution of ARPES spectra reported by Kurokawa et al., *Nat. Commun.* 14 (2023) suggest a smooth evolution of magnetic polarons into the ubiquitous Fermi arcs with suppressed spectral weight outside the magnetic Brillouin zone.

To conclude, there is numerical and experimental evidence that in the underdoped Fermi-Hubbard model the hole-like quasiparticle is attached to strings of length $\ell \neq 0$. This is fully consistent with ARPES measurements since the spectral weight and the character of the quasiparticles can change with doping while maintaining polaronic spin-charge correlations relevant for our Feshbach mechanism.

Reviewer:

Last but not least, I am also quite worried that this paper has too much overlap with a recent article published by some of the Authors as [Homeier et al., *PRB* 109, 125135 (2024)].

Our reply:

We appreciate the referee's comparison to the recent PRB. The PRB paper is dedicated to technical calculations and to improvements of the magnetic polaron formalism in terms of a truncated basis approach. However, the main text of this paper has little to do with the more general Feshbach mechanism proposed in this manuscript. Instead the technical calculations are motivated by the current manuscript and hence we provide the reader with a brief summary in the Introduction.

Reviewer:

That is why I am still quite reluctant in recommending this manuscript for publication in *Nature Communications*.

Our reply:

We hope our reply now, along with our changes to the manuscript, also clarifies the relevance of our work in the finite doping regime and that the features seen in ARPES spectra at low doping are fully consistent with the parton and polaron picture used in our study. In this vein, we would be delighted for a recommendation to publish our manuscript in *Nature Communications*.

Response to the Reviewer #3

Reviewer:

My original report was the least critical of the three. My main issue was regarding to what extent experimental data supports the conclusion that the Feshbach resonance mechanism provides the best interpretation of the data. In the authors' response they share some preliminary results comparing SCPA and DMRG results for the two-hole ARPES spectra. While interesting, I don't think this really answers my question, so the paper still strikes me as rather speculative. In that regard, I believe this manuscript might be more suitable for a journal such as Phys. Rev. B, although I would not object to publication in Nature. As I had noted before, it is quite clear and well-written and the figures are very well-done. Furthermore, there is significant new material here vis-a-vis earlier work on this mechanism as a driver of high temperature superconductivity.

Our reply:

We thank the reviewer again for the nice words on the presentation of our manuscript and for the acknowledgments about the significance of our study. Again, we would like to emphasize that we believe that *Nature Communications* is an ideal venue for our work in order to bring it to the attention to a broader readership of experimentalists and theorists and test our hypothesis in the near future with in either experiments on solids, cold atoms or in numerical studies. We appreciate the reviewer shares this opinion partially and does not *object to publication in Nature*.

In the revised version of our manuscript, we have further included additional details about the parton picture and we clarify that our mechanism only requires short-range polaronic correlations; hence we believe that in the most important regime of the underdoped cuprates the Feshbach mechanism should be valid. Additionally, we have now dedicated an entire section to Feshbach mechanisms in other solid state materials, which have recently created attention in the field.

With these changes, we hope the Reviewer will recommend our manuscript for publication in *Nature Communications*.